# Single-cell total-RNA profiling unveils regulatory hubs of transcription factors

Yichi Niu [1,2,7], Jiayi Luo[1,3,7] & Chenghang Zong [1,2,3,4,5,6] ✉

Recent development of RNA velocity uses master equations to establish the kinetics of the life cycle of RNAs from unspliced RNA to spliced RNA (i.e., mature RNA) to degradation. To feed this kinetic analysis, simultaneous measurement of unspliced RNA and spliced RNA in single cells is greatly desired. However, the majority of single-cell RNA-seq chemistry primarily captures mature RNA species to measure gene expressions. Here, we develop a one-step total-RNA chemistry-based single-cell RNA-seq method: snapTotal-seq. We benchmark this method with multiple single-cell RNA-seq assays in their performance in kinetic analysis of cell cycle by RNA velocity. Next, with LASSO regression between transcription factors, we identify the critical regulatory hubs mediating the cell cycle dynamics. We also apply snapTotal-seq to profile the oncogene-induced senescence and identify the key regulatory hubs governing the entry of senescence. Furthermore, from the comparative analysis of unspliced RNA and spliced RNA, we identify a significant portion of genes whose expression changes occur in spliced RNA but not to the same degree in unspliced RNA, indicating these gene expression changes are mainly controlled by post-transcriptional regulation. Overall, we demonstrate that snapTotal-seq can provide enriched information about gene regulation, especially during the transition between cell states.

The rapid development of single-cell RNA-seq (scRNA-seq) has enabled large-scale characterization of different cell states, which provides valuable information about the changes that occurred during the cell-state transitions, such as development and differentiation processes. Following this characterization, it is desired to decode the underlying regulatory mechanisms. We reason that this regulatory information could be derived based on the comparative analysis between unspliced RNAs and spliced RNAs (i.e., mature RNAs) detected in single cells. Here, our study shows that effective detection of unspliced transcripts by total-RNA-based scRNA-seq methods indeed renders them significant advantages for kinetic and regulatory analyses, besides the ability to detect all RNA biotypes (protein coding, long non-coding and short non-coding RNAs). It is worth noting that the majority of scRNA-seq methods, including the high-throughput platform of 10× chromium, primarily capture mature RNA with oligo-dT primers[1–5], making them less ideal for the comparative analysis due to inefficient detection of unspliced RNA species[6].

So far, three main single-cell total-RNA methods have been developed in recent years, including MATQ-seq[7] by Sheng et al. in ref. 7, Smart-seq-total method[8] by Isakova et al. in ref. 8, and VASA-seq by Salmen et al. in ref. 9. Total-RNA-seq chemistry has also been applied to profile the transcriptome of individual neuronal synaptosomes[10]. Despite the advantage of the single-cell total-RNA sequencing approach for RNA velocity analysis[11,12], the chemistries of current total-RNA-based methods are generally more complicated than mature RNA-based methods, especially in comparison to

[1]Department of Molecular and Human Genetics, Houston, TX, USA. [2]Genetics & Genomics Program, Houston, TX, USA. [3]Cancer and Cell Biology Program, Houston, TX, USA. [4]Integrative Molecular and Biomedical Sciences Program, Houston, TX, USA. [5]Dan L Duncan Comprehensive Cancer Center, Houston, TX, USA. [6]McNair Medical Institute, Baylor College of Medicine, Houston, TX, USA. [7]These authors contributed equally: Yichi Niu, Jiayi Luo. ✉ e-mail: chenghang.zong@bcm.edu

SMART-seq chemistry[4,13,14]. Here, by combining multiple annealing chemistry allowed by MALBAC primers[15] and the template-switching chemistry[4,14,16] (Supplementary Fig. 1), we develop a one-step single-cell total-RNA-seq chemistry. We refer to this assay as snapTotal-seq, which can be easily implemented on liquid-handling platforms.

Next, we benchmark the performance of snapTotal-seq, SMART-seq3, CEL-Seq2, SMART-seq-total, and VASA-seq in their ability to capture the cell cycle dynamics through RNA velocity analysis. As a result, in comparison to mature RNA-based methods, total-RNA-based methods demonstrate substantial improvement in recapitulating the transcriptional dynamics of the cell cycle, with snapTotal-seq achieving the best performance.

With the trajectory data from RNA velocity, we show that by LASSO regression of the unspliced RNA expressions of a transcription factor (TF) against the spliced RNA expressions of the rest of TFs, we identify the important TF hubs that mediate the cell-state transitions during the cell cycle. Furthermore, the comparative analysis between unspliced RNA and spliced RNA expression also reveals the substantial role of post-transcriptional regulation in the expression of a significant portion of cell cycle genes.

Besides the cell cycle, we further expand the application of snapTotal-seq to investigate the regulatory network in oncogene-induced senescence (OIS). As a result, we successfully identify the key regulatory components that orchestrate cell-state transition during OIS. Overall, these findings underscore the ability of snapTotal-seq to characterize the transcription/splicing/degradation dynamics of RNAs and, with greater significance, to identify the key transcription hubs driving cell-state transitions.

## Results

### Chemistry of snapTotal-seq

The overall chemistry of snapTotal-seq is shown in Fig. 1a and Supplementary Fig. 2. After the lysis of single cells; we used MALBAC primers[15] to initiate the reverse transcription at low temperature, which allows the random but efficient primer binding to multiple sites on RNAs, therefore warranting the total-RNA detection ability. Next, we gradually ramped up the temperature to promote the reverse transcription. Once the reverse transcription stopped at the sites with difficult secondary structures or reached the 5′ end of RNA, template switching then occurred with the existing MALBAC primers serving as the template-switching oligos. In total, we performed ten cycles of multiple annealing and extension (without melting steps), during which cDNA amplicons were successfully generated by template switching. With the special design of MALBAC primers, the primer dimer is not detectable in the final product even when the annealing steps reach as low as 8 °C. After cDNA amplification, cell-specific barcodes were introduced to the amplified products by a double-strand conversion step. Next, the cells with different barcodes were pooled together for the library construction (Supplementary Fig. 2). It is worth emphasizing that all the reads in our method have UMI sequences.

We first performed the cross-species experiment using HEK293T cells and 3T3 mouse cells. We observed that the crosstalk between the two species was rare (Supplementary Fig. 3a). Next, we performed two technical replicates to show that technical variations were minimal as the single-cell data of different batches were completely overlapped in the principal component analysis (PCA) (Supplementary Fig. 3b). With the total-RNA approach, our method is able to detect different classes of non-coding RNA alongside protein-coding genes (Supplementary Fig. 3c). Overall, we were able to detect $10,400 \pm 500$ (mean ± sd) genes based on exon reads and $10,400 \pm 900$ (mean ± sd) genes based on intronic reads with -1 million uniquely mapped reads per cell (Fig. 1b), which confirms the effective capture of both spliced RNA and unspliced RNA by snapTotal-seq.

### Benchmarking comparison of single-cell RNA-seq methods in RNA velocity analysis

Besides the ability to efficiently detect different RNA biotypes, the primary advantage of total-RNA-based methods lies in their ability to provide the data that facilitate the modeling of gene expression processes, from transcription to splicing and RNA decay, as described by the master equations of the RNA velocity[11,12]. We first applied RNA velocity analysis to the snapTotal-seq data. As shown in Fig. 1c–e, we observed that the velocities clearly captured the dynamic transitions between different cell cycle phases (G2/M to >G1 to >S to >G2/M) with high confidence scores ($0.93 \pm 0.073$, Supplementary Fig. 3d).

To verify the result of RNA velocity analysis, we applied another algorithm: reCAT[17] to analyze the cell cycle dynamics using the same data. Different from ab initio based analysis by RNA velocity, reCAT determines the cell cycle dynamics based on the known cell cycle genes. The pseudotime derived from reCAT is shown in Supplementary Fig. 3e. And the latent time from RNA velocity and the pseudotime from reCAT were highly correlated with Pearson's coefficient larger than 0.9 (Supplementary Fig. 3f).

To further examine the cell–cell heterogeneity unveiled by our method, we then performed dimensionality reduction by using UMAP. Interestingly, through the unsupervised clustering, we identified five distinct cell cycle sub-stages in our data, as evidenced by the expression profiles of a set of well-known cell cycle marker genes (Supplementary Fig. 4a–d). We also observed the differential expression of non-coding RNAs across different cell cycle sub-stages (Supplementary Fig. 4e). The dynamic transitions between these sub-stages were also recapitulated by RNA velocity analysis (Supplementary Fig. 4f). These results further demonstrate that our method can effectively recapitulate the intrinsic dynamic heterogeneity.

Next, we performed a benchmark comparison between snapTotal-seq and four scRNA-seq methods: Smart-seq3[4], CEL-Seq2[5], VASA-seq[9] (plate version), and Smart-seq-total[8]. Among them, Smart-seq3 and CEL-Seq2 employ oligo-dT-based reverse transcription strategy, while VASA-seq and Smart-seq-total are two recently developed total-RNA-based scRNA-seq methods. First, we observed that the gene detection rate of snapTotal-seq is similar to VASA-seq based on either exon reads or intronic reads. And they are substantially higher than the two dT-based methods (Supplementary Fig. 5a, b). Consistently, compared to dT-based methods, both snapTotal-seq and VASA-seq exhibit a more uniform read distribution over the gene body (Supplementary Fig. 5c), and have a higher proportion of intronic reads (Supplementary Fig. 5d). Interestingly, we noticed Smart-seq-total has a low gene detection with either exon or intron reads, which is potentially due to the dominant detection of a few genes, specifically *RN7SK*, *RMRP*, *RPS29*, and *KIAA0907* in their data[8] (Supplementary Fig. 5e). Due to this significant detection difference, we excluded Smart-seq-total from the downstream benchmark analyses.

Next, we investigated the technical features of our method in detail. First, we showed that the improvement in gene detection by both snapTotal-seq and VASA-seq was consistently observed in different RNA biotypes (Supplementary Fig. 5f), suggesting that the total-RNA strategy improved the gene detection across all categories of genes. Furthermore, we showed that both total-RNA-based methods significantly improve the capturing efficiency of the transcripts with moderate or low expression levels at both exon and intron levels (Supplementary Fig. 5g). At last, similar to VASA-seq, snapTotal-seq exhibits a clear correlation between the gene detection rate and RNA length (Supplementary Fig. 5h), suggesting it is a common feature of total-RNA-based approaches. In dT-based methods, although only a slight bias was observed in exon reads, a clear dependence was noted in intron reads, given that the capture of unspliced RNA by dT-based methods relies on the binding of oligo-dT primer to the polyA sequence in introns[11]. Furthermore, it is worth noting that the overall capture efficiency for short RNA in our method (81.4% for exon reads,

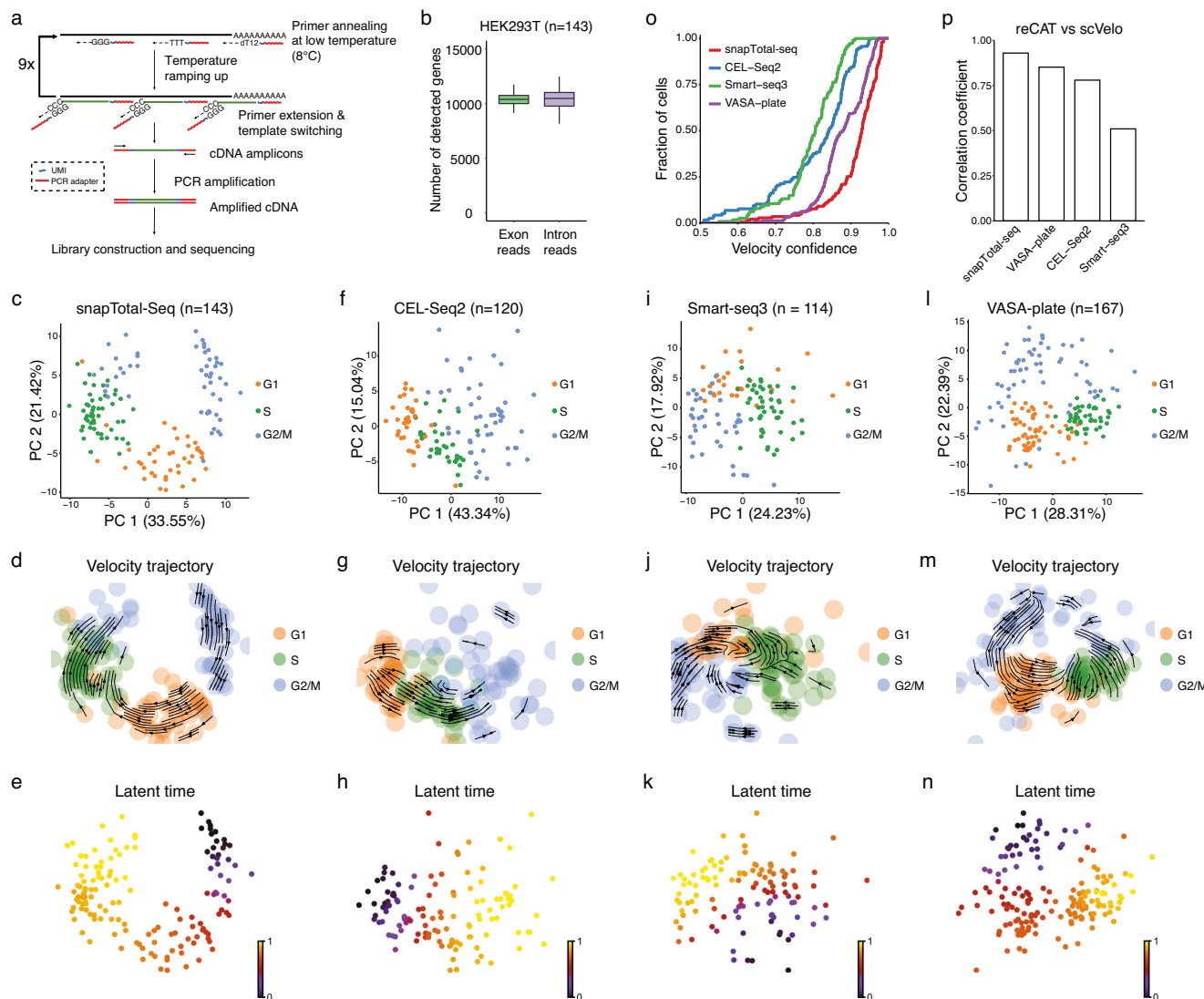

**Fig. 1 | Development of snapTotal-seq and benchmark analysis. a** Schematic of snapTotal-seq. **b** The number of detected genes by exon reads, or intron reads in single HEK293T cells by snapTotal-seq. Center lines show median, box limits show the upper and lower quartiles, and whiskers show the 1.5× interquartile range (IQR). **c** PCA plot of the HEK293T cells. The cells are colored by their corresponding cell cycle phases. **d** The velocity trajectory projected by RNA velocity analysis. **e** Latent time inferred by RNA velocity. **f** PCA plot of the HEK293T cells sequenced by CEL-Seq2. **g** The projected velocity trajectory of the HEK293T cells sequenced by CEL-Seq2. **h** Latent time inferred by RNA velocity for CEL-Seq2. **i** PCA plot of the HEK293T cells sequenced by Smart-seq3. **j** The projected velocity trajectory of the HEK293T cells sequenced by Smart-seq3. **k** Latent time inferred by RNA velocity for Smart-seq3. **l** PCA plot of the HEK293T cells sequenced by VASA-plate. **m** The projected velocity trajectory of the HEK293T cells sequenced by VASA-plate. **n** Latent time inferred by RNA velocity for VASA-plate. **o** The velocity confidence scores of the HEK293T cells sequenced by different methods. **p** The correlation coefficients between the reCAT based cell cycle trajectory and the RNA velocity based cell cycle trajectory achieved by different methods. Source data are provided as a Source Data file. In (**b**, **c**, **f**, **i**, and **l**), "n" represents cell number.

68.2% for intron reads) is also higher than the capture efficiency of both dT-based methods (CEL-Seq2: 75.5% for exon reads and 53.8% for intron reads; Smart-seq3: 74.5% for exon reads and 45.7% for intron reads) for the same class of RNA (Supplementary Fig. 5h).

Next, we performed PCA analysis and RNA velocity analysis on each dataset. We observed that the connections between different cell cycle states were not well positioned for both dT-based methods in the PCA plots (Fig. 1f–k). In contrast, both snapTotal-seq (Fig. 1c–e) and VASA-seq (Fig. 1l–n) showed a clear circular distribution describing different cell cycle phases. As a result, the velocity confidence scores of Smart-seq3 and CEL-Seq2 were significantly lower than those of snapTotal-seq and VASA-seq. We also noticed that snapTotal-seq has a better confidence score than VASA-seq (Fig. 1o).

Next, we also employed reCAT analysis to infer the cell cycle trajectory based on the expression of known cell cycle genes

(Supplementary Figs. 3e and 6). As a result, our method achieved the highest correlation coefficient between the results of the unsupervised and supervised analysis (Fig. 1p), confirming the consistent performance of the two independent analyses using snapTotal-seq data.

At last, we performed dimensionality reduction on these datasets to investigate whether additional information could be unveiled in high dimensions. As a result, despite the enhanced separation between different cell cycle stages with the inclusion of more dimensions, the cell cycle sub-stages were not identified in these datasets with unsupervised clustering (Supplementary Fig. 7), in contrast to the observations in snapTotal-seq data (Supplementary Fig. 4). Based on the systematic comparison described above, we concluded that snapTotal-seq outperformed other methods in characterizing the dynamic heterogeneity and the underlying gene expression kinetics.

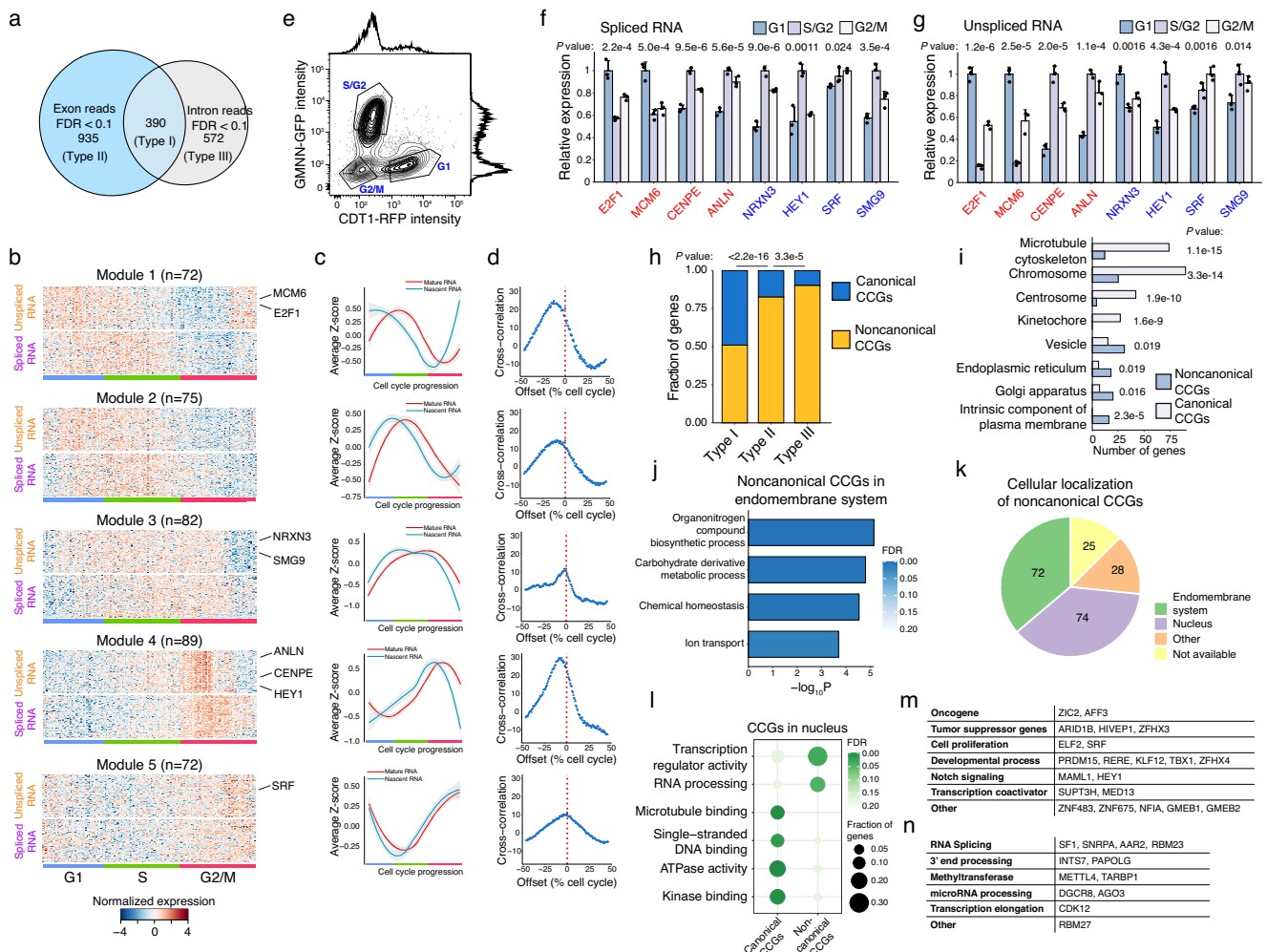

**Fig. 2 | Functional analysis on Type I cell cycle genes (CCGs). a** Differentially expressed genes identified at spliced RNA or unspliced RNA level along the cell cycle in HEK293T cells. **b** Gene expression heatmap of Type I CCGs along the cell cycle. Genes were clustered into five kinetic modules using k-means clustering algorithm based on their transcriptional dynamics along the cell cycle. **c** The smoothed gene expression curves along the cell cycle for each kinetic modules. The smoothed curves were derived by using Loess function. Center line represents the average, and shade represents the 0.95 confidence interval. **d** Cross-correlation between the expression of unspliced RNA and spliced RNA. Cross-correlation was calculated using numpy.correlate function in Python. **e** Flow cytometry analysis on HEK293T-FUCCI cells. Gating strategies are provided in Supplementary Fig. 23. qRT-PCR validation of the differential expression of eight CCGs at spliced RNA level (**f**) and at unspliced RNA level (**g**) in different cell cycle phases. Canonical CCGs are labeled in red, and noncanonical CCGs are labeled in blue. Three biological replicates were performed for each measurement. Mean and standard deviation were shown. One-way ANOVA was performed. Source data are provided as a Source Data file. **h** The proportions of noncanonical CCGs in three types of CCGs. The list of canonical CCGs is compiled by including the cell cycle pathway in Gene Ontology database, the cell cycle pathway in Reactome database, and the gene list from Cyclebase. **i** Differential enrichment in different cellular compartments between canonical CCGs and noncanonical CCGs. **h, i** Two-sided Fisher's exact tests were performed. Source data are provided as a Source Data file. **j** Gene Ontology (GO) enrichment of noncanonical CCGs localized in endomembrane system. One-sided Fisher's exact test was performed and FDR was calculated using Benjamini–Hochberg procedure. **k** Cellular localization of all noncanonical CCGs. **l** GO enrichment of canonical CCGs and noncanonical CCGs localized in the nucleus. One-sided Fisher's exact test was performed and FDR was calculated using Benjamini–Hochberg procedure. **m** Functional classification of the transcription factors belonging to noncanonical CCGs. **n** Functional classification of the RNA processing factors belonging to noncanonical CCGs.

## Comparative analysis between unspliced and spliced RNAs identified three types of cell cycle genes (CCGs)

Based on the constructed cell cycle trajectory as described above, we then performed the trajectory-based differential gene expression analyses on exon reads (representing spliced RNA) and intron reads (representing unspliced RNA) using tradeSeq[18], respectively. We identified 1325 genes with significant changes at the spliced RNA level (FDR < 0.1) and 962 genes with significant changes at the unspliced RNA level (FDR < 0.1) along the cell cycle (Fig. 2a and Supplementary Data 1).

We noticed that there are 390 genes that showed significant changes in both unspliced and spliced RNA, which we denoted as Type I CCGs, and 935 genes that showed only significant changes in

spliced RNA, which we denoted as Type II CCGs, and 572 genes that showed only significant changes in unspliced RNA, which we denoted as Type III CCGs. For Type II CCGs, the lack of significant changes in unspliced RNA suggests that gene expression changes that occurred in the spliced RNAs are mainly contributed by post-transcriptional regulation (Supplementary Fig. 8a, b and Supplementary Data 2). In comparison, gene expression changes in Type I CCGs are mainly contributed by transcriptional regulation. For a large number of genes in the Type III CCGs, it is interesting to see the transcriptional variations that occurred to these genes are effectively buffered out at the spliced RNA level, likely also through post-transcriptional regulation-based mechanisms, which is worth future investigation.

## Identification of five kinetic modules and noncanonical genes in Type I CCGs

For Type I CCGs, we identified five kinetic modules (Fig. 2b; Supplementary Fig. 8c–g and Supplementary Data 3). As shown in Fig. 2b, c, a clear time delay occurs between the changes in unspliced RNA and spliced RNA, which mainly corresponds to the splicing process. We quantified the coupling between the dynamics of unspliced RNA and spliced RNA by the cross-correlation between two expression curves (Fig. 2d). The close coupling confirms that the changes in the expression of these genes mainly originated from the regulation at transcription step.

To validate the detected differentially expressed genes (DEGs) by snapTotal-seq along the cell cycle, we used HEK293T cells with the expression of FUCCI markers[19] to collect the cells at G1, S/G2 and G2/M phases (Fig. 2e). We then randomly chose eight genes from the detected Type I CCGs and performed qRT-PCR to measure their gene expression levels. As a result, the differential expression of these genes at different cell cycle stages was confirmed for both unspliced RNA and spliced RNA (Fig. 2f, g).

Besides the genes with known functions in cell cycle regulation (i.e., canonical CCGs), we observed that over 50% of Type I CCGs (199 out of 390) had not been associated with cell cycle regulation previously (here we refer to them as noncanonical CCGs) (Fig. 2h and Supplementary Fig. 8h). In contrast to canonical CCGs, which are significantly enriched with mitosis related structures, the noncanonical CCGs are significantly enriched in the endomembrane system, including endoplasmic reticulum, Golgi apparatus, and intrinsic component of plasma membrane (Fig. 2I, j). Functional enrichment shows that these genes are involved in the synthesis of organonitrogen compounds, metabolism of carbohydrate derivatives, ion transportation, and maintenance of chemical homeostasis (Fig. 2k).

Next, the second major subset of noncanonical CCGs (74 out of 199) reside in the nucleus (Fig. 2j). In contrast to the canonical CCGs in the nucleus that are enriched with microtubule binding, single-strand DNA binding and ATPase activity, the noncanonical CCGs are enriched in transcriptional regulators and RNA processing factors (Fig. 2l). The related pathways of these noncanonical CCGs are shown in Fig. 2m for transcriptional regulators and Fig. 2n for RNA processing factors. The identification of the abundant noncanonical cell cycle genes directly shows the advantage of our method in transcriptional kinetics analysis.

## Identification of TF hubs regulating the five kinetic modules along the cell cycle

Next, we sought to identify the transcription factors (TFs) that could govern the five kinetic modules observed in Type I CCGs. To do so, we utilized LASSO regression[20] to identify the associations between the spliced RNA changes of TFs and the unspliced RNA changes of the rest of the genes in different modules (Fig. 3a). It is worth noting that, in comparison to oligo-dT-based methods, the simultaneous detection of both unspliced RNA and spliced RNA by our method allows us to directly link the expression of TFs (spliced RNA levels) to the transcriptional dynamics of their target genes (unspliced RNA levels) through this analysis. As a result, we identified 23 TFs with their potential downstream genes being significantly enriched in Type I kinetic modules (Fig. 3b). Furthermore, based on the transcriptional activities of these 23 TFs (i.e., the changes in the unspliced RNA levels of their target genes), we successfully identified four TF hubs (Fig. 3c). The correlation matrix between these hubs is shown in Fig. 3d.

Next, to identify the potential connections between these 23 TFs, we performed the LASSO regression between the changes in the unspliced RNA of a TF and the changes in the spliced RNA of the rest of the expressed TFs using LASSO regression. This analysis produces the TF association network as shown in Fig. 3e. Interestingly, we noticed that Hub 3 (light blue gene blocks), while only composed of 3 genes,

was located in the middle of Hub 2 (yellow gene blocks) and Hub 4 (cyan gene blocks), and was connected to the genes in both Hub 2 and Hub 4, indicating that Hub 3 plays important roles in mediating the transition from the G1/S to G2/M state.

## Validation of TF regulations based on ChIP-seq data and motif analysis

From the LASSO regression inferred TF hubs, we next evaluate the direct regulation based on the published ChIP-seq datasets[21] and motif analysis[22] ("Methods"). Among the candidate TFs, the ChIP-seq data were unavailable for 12 of them. For the 11 TFs with ChIP-seq data, we identified 6 TFs (E2F1, E2F2, KLF11, FOXM1, NFYC, and MYBL2) whose downstream genes showed significant enrichment with the binding targets identified by ChIP-seq (FDR < 0.05, Supplementary Fig. 8i) or displayed a significant enrichment of the corresponding motif sequence at their promoter regions (normalized enrichment score (NES) > 3, Supplementary Data 4), confirming the direct regulatory relationships between these TFs and their downstream genes identified by LASSO regression.

Among the 6 TFs verified by ChIP-seq data, E2F1, E2F2, and KLF11 TFs have been known for their roles in regulating G1/S transition[23,24], and consistently, the transcriptional activities of the associated gene modules were clearly increased in G1 and early S phases (Fig. 3f). The regulation relation between KLF11, E2F1, and E2F2 is shown in Fig. 3g–i. Next, for both FOXM1 and NFYC, their associated gene modules were highly activated in late S and early G2 phases (Fig. 3f). Consistent with this observation, FOXM1 is the well-known master regulator of G2/M phase[23,25,26]. Next, we observed that the MYBL2, another major regulator of G2/M gene expression[27,28], was clearly activated prior to the activation of FOXM1 module (Fig. 3f), implying that MYBL2 functions as an intermediate TF that could play important roles in the transition from G1/S to G2/M state. It is worth noting that MYBL2 is located in Hub 3. The detailed regulation relation between MYBL2 and KLF11, E2F1, E2F2, and FOXM1 are shown in Fig. 3j, k.

Overall, we observed that the G1/S TFs formed a positive feedback loop to promote G1/S phase gene expression (Fig. 3l). Consistent with previous findings by Skotheim et al.[29], the positive feedback loops ensure the stability of G1/S state. Interestingly, our data unveiled that all of these G1/S TFs positively regulated the expression of MYBL2 (Fig. 3j, l). The upregulated MYBL2 subsequently activated the expression of FOXM1 (Fig. 3k, l), leading to the activation of G2/M phase gene expression. This observation shows the pivotal role of MYBL2 in driving the cell-state transition from G1/S toward G2/M. To further verify the generality of this observation, we sequenced another two cell lines (U2OS and HPNE) using snapTotal-seq, and similar results were observed (Supplementary Figs. 9, 10).

In summary, our results demonstrated the ability of snapTotal-seq to recapitulate the TF regulatory network that drives the cell-state transition. For the 5 TFs (5 out of the 11 TFs with ChIP-seq data) that lack significant enrichment in their ChIP-seq data, it could be caused by either the binding targets revealed by ChIP-seq being cell line specific or the identified downstream genes being indirect targets of these TFs. Future studies are required to further investigate the regulatory roles of these TFs in cell cycle progression.

## TF analysis with non-parametric approach of GENIE3

To further verify the results obtained through LASSO regression, we employed GENIE3[22,30], a non-parametric method, as an alternative approach to infer TF modules based on the covariation between the spliced and unspliced RNA of different genes. As a result, 12 out of 23 TF modules identified by LASSO regression were also captured by GENIE3, which includes all of six core TFs modules verified by ChIP-seq data and motif enrichment analysis (Supplementary Fig. 11a–c). Within these shared TF modules, 295 out of 439 TF-gene links identified by LASSO-based analysis were confirmed by GENIE3 (Supplementary

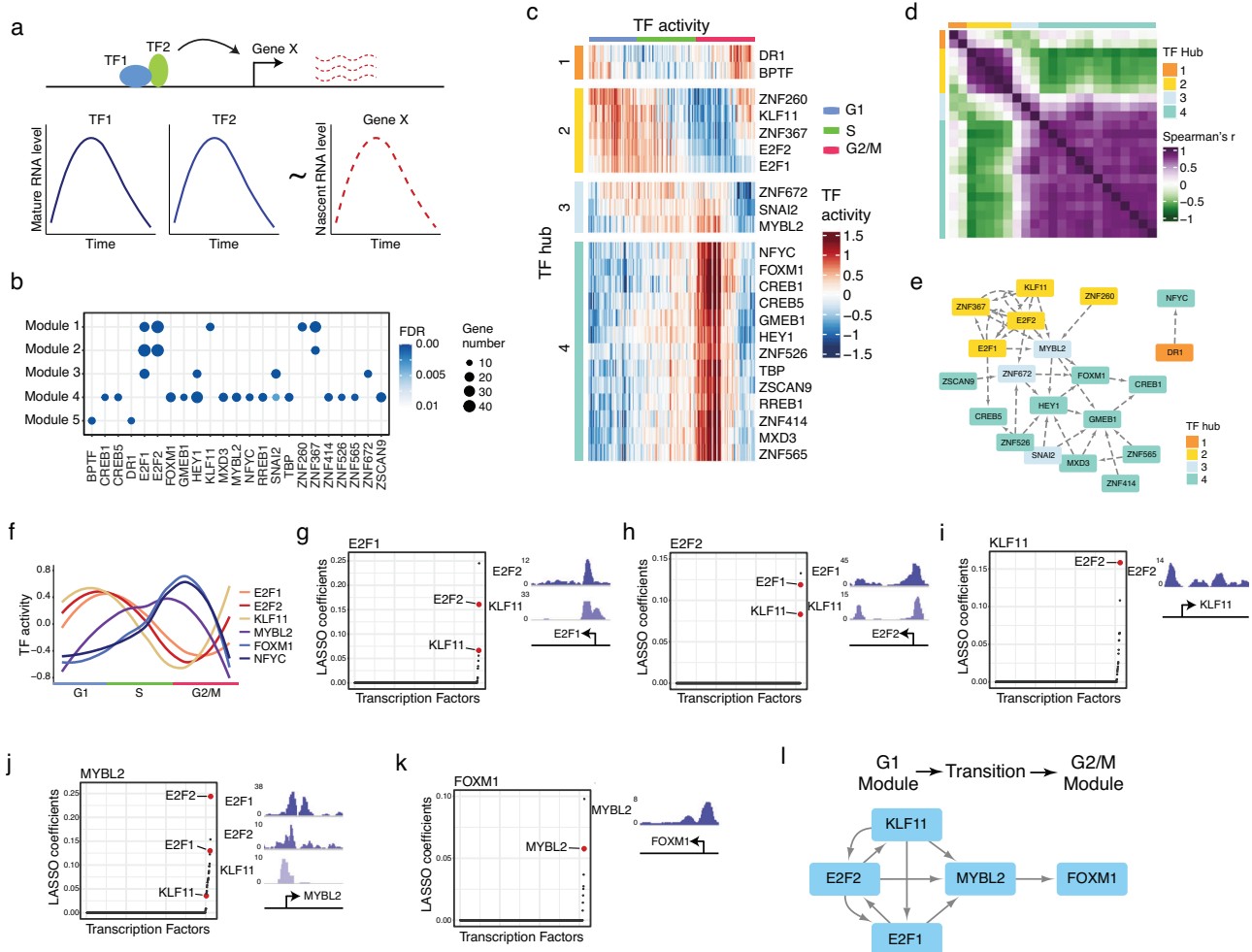

**Fig. 3 | Reconstructing transcription factor (TF) regulatory network of the cell cycle. a** Schematic outlining the approach used to establish the regulatory links between TFs and their target genes. **b** TFs whose associated genes are significantly enriched in Type I kinetic modules. Statistical significance was calculated using one-sided Fisher's exact test and FDR was calculated using Benjamini–Hochberg procedure. **c** Heatmap of the activities of the TFs with significant enrichment with Type I kinetic modules. **d** Heatmap of the correlation coefficients between the activities of each pair of TFs. **e** TF association network established based on the TF–TF links identified with LASSO regression. We noticed that three TFs (BPTF, RREB1, TBP1) were missing in this TF network. The potential functions of these three TFs are discussed in Supplementary Note 1. **f** The activities of the verified TFs along the cell cycle. The smoothed curves were derived by using Loess function. **g–k** Establish the regulatory links between different TFs by correlating the changes in the unspliced RNA of the TF of interest with the changes in the spliced RNA of the other TFs. The direct regulatory links (colored in red) were identified by ChIP-seq verification. The corresponding ChIP-seq peaks were plotted on the right. The list of all the TFs with LASSO coefficients is provided in Supplementary Data 5. **l** TF regulatory network established based on the direct regulatory links between different TFs.

Fig. 11f). In addition, the TF modules and the TF-gene links detected by both approaches exhibit significantly higher coefficients/weight than those detected by only one approach (Supplementary Fig. 11c–h), suggesting that most of the TF-gene links with high confidence have been captured in LASSO-based analysis. At last, we examined the regulatory relationships between different TFs using GENIE3, and, consequently, all the regulatory relationships among the core TFs described above were also verified by GENIE3-based analysis (Supplementary Fig. 12).

## Identification of noncanonical CCGs under post-transcriptional regulation

Type II CCGs were featured by their significant changes at the spliced RNA level but not at the unspliced RNA level (Supplementary Fig. 5a, b and Supplementary Fig. 13), indicating the involvement of post-transcriptional regulation of these genes. Based on their dynamic changes along the cell cycle, we also identified five kinetic modules (Fig. 4a, b and Supplementary Data 3). The cross-correlation analysis shows that the coupling between the gene expression changes in unspliced RNA and spliced RNA is not as significant as Type I CCGs (Figs. 2d and 4c).

Gene ontology (GO) enrichment analysis was then carried out on each kinetic module. Module 2 has significant enrichment with DNA replication and repair pathways. Modules 1, 4, and 5 were significantly enriched with pathways that are not directly associated with cell cycle progression (Fig. 4d). These pathways include RNA metabolism, RNA processing, vesicle transportation, etc. These results suggest that the activities of many biological processes are coordinated with cell cycle progression by post-transcriptional regulations.

To evaluate the important roles of the noncanonical Type II CCGs in the cell cycle, we examined the essentiality of these genes by using the genetic screening datasets from Project Achilles[31]. As a result, we found that the pan-essential genes were significantly overrepresented among the noncanonical CCGs in Modules 1, 4, and 5 (Fig. 4e). This observation indicates that the functions of these post-transcriptionally regulated kinetic modules are also important for cell proliferation.

Next, we explored whether these genes play critical roles in cancer progression since cancer is a disease of dysregulation of the cell cycle.

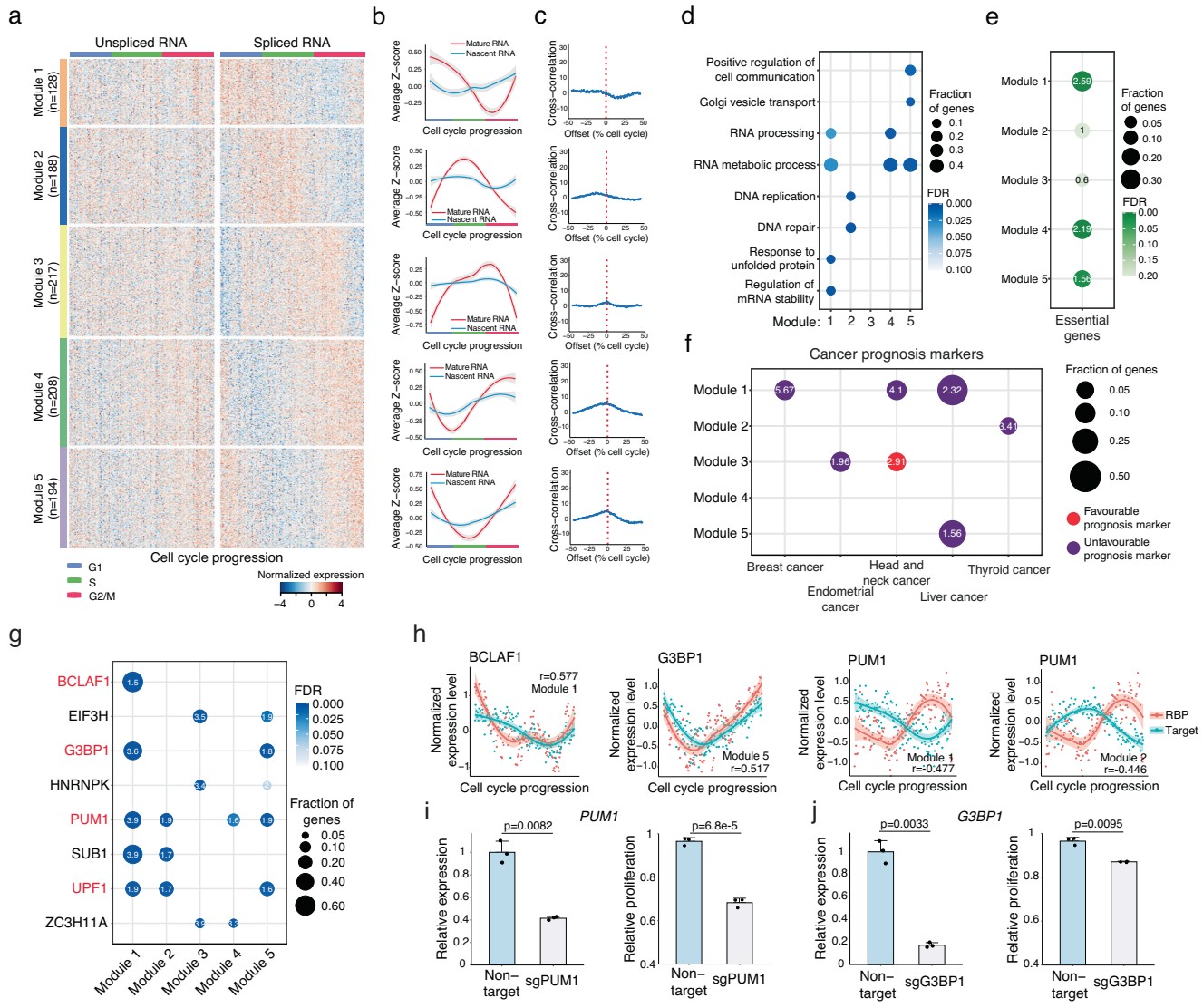

**Fig. 4 | Post-transcriptional regulation in Type II cell cycle genes (CCGs). a** Gene expression heatmap of Type II CCGs at spliced RNA and unspliced RNA levels along the cell cycle. Genes were clustered into five kinetic modules using k-means clustering algorithm based on the expression patterns of their spliced RNA along the cell cycle. **b** The smoothed gene expression curves along the cell cycle for five kinetic modules. **c** The cross-correlation between the expression curves of unspliced RNA and spliced RNA. The cross-correlation was calculated by using numpy.correlate function in Python. **d** GO enrichment in five kinetic modules of Type II CCGs. **e** Significant enrichment of pan-essential genes in different gene modules. Canonical CCGs are excluded in this analysis. **f** Significant enrichment (FDR < 0.1) of cancer prognostic markers in different gene modules. Canonical CCGs are excluded in this analysis. **g** Significant enrichment (FDR < 0.1) of the target genes of the RNA binding proteins in Type II CCGs. **h** Gene expression dynamics of *BCLAF1*, *G3BP1*, *PUM1* and their target gene modules along the cell cycle. Only the

gene modules whose expression patterns were significantly correlated (r > 0.4 or r < −0.4) with the corresponding RBPs were considered as the target gene modules. **i** Cell proliferation was significantly decreased in *PUM1* knockdown cells compared to the cells with non-target sgRNA. **j** Cell proliferation was significantly decreased in *G3BP1* knockdown cells compared to the cells with non-target sgRNA. For the gene expression curves in (**b**, **h**), the center line represents the average and the shade represents 0.95 confidence interval. The smoothed curves were derived by using Loess function. For the enrichment analyses in (**e**–**g**), one-sided Fisher's exact tests were performed and FDR was calculated using Benjamini−Hochberg procedure. The enrichment folds were shown in the figure. For the bar plots in (**i**, **j**), three biological replicates were performed for each measurement. Mean and standard deviation were shown. Two-sided student's t-tests were performed and *P* values were shown in the figure. Source data are provided as a Source Data file.

Interestingly, we found that except for Module 4, the noncanonical CCGs in the rest of the modules were significantly enriched with the prognostic markers of different cancer types[32], further indicating that cancer cells likely need to alter post-transcriptional regulations to fit with the abnormal cell proliferation (Fig. 4f).

At last, to rule out the possibility that the noncanonical CCGs identified in our study might be biasly detected by snapTotal-seq, we examined the changes of their gene expression across the cell cycle in VASA-seq data. As a result, despite the noted differences in culture conditions between VASA-seq study and ours, similar patterns of cell

cycle dependent gene expression changes were consistently observed for these noncanonical CCGs in VASA-seq data, which verified our findings (Supplementary Note 2 and Supplementary Fig. 14).

## Multiple post-transcriptional regulation mechanisms underlying Type II CCGs

Next, we investigated the potential mechanisms that drive the differential expression of Type II CCGs. Considering that RNA binding proteins (RBPs) have been reported as a class of proteins regulating the fate of RNA at different post-transcriptional processing steps[33–35], we

examined their potential roles in regulating the differential expression of Type II CCGs. Among the known 150 RBPs whose binding targets have been thoroughly examined[36], 22 RBPs were differentially expressed along the cell cycle in our data.

To evaluate their functions in cell cycle progression, we then compared their binding targets to Type II CCGs. As a result, we identified eight RBPs whose binding targets were significantly enriched in at least one Type II CCG module (Fig. 4g). It is worth pointing out that four out of these eight RBPs (*UPF1*, *BCLAF1*, *PUM1*, and *G3BP1*) have already been reported to regulate RNA stability and decay[36]. To further substantiate the potential regulatory roles of these RBPs, we examined the correlation between the expression patterns of the four RBPs and their target genes. For *BCLAF1* and *G3BP1*, known to enhance RNA stability[37,38], we successfully identified the target gene groups exhibiting a positive correlation with the expression patterns of these two RBPs (Fig. 4h). For *PUM1*, which has been reported to promote RNA decay[39–41], we identified its target gene groups showing a negative correlation with its own expression pattern (Fig. 4h).

Next, to validate our findings, we knocked down two non-essential genes: *PUM1* and *G3BP1* (Fig. 4i, j and Supplementary Fig. 15a). We did not test the knockouts of *BCLAF1* since it has been classified as a pan-essential gene[31]. As a result, we observed that the cell growth was significantly decreased in *PUM1* knockdown cells (Fig. 4i), which suggests that *PUM1* indeed plays a critical role in regulating cell proliferation in the HEK293T cell line. The knockdown of *G3BP1* also led to decreased cell proliferation with statistical significance (Fig. 4j).

It is also worth pointing out that we identified 14 genes whose binding targets were not significantly enriched in Type II CCGs. Interestingly, these genes are mainly composed of splicing factors, rRNA processing factors, and miRNA processing factors (Supplementary Fig. 15b). For the splicing factors, the lack of target enrichment in Type II CCGs suggests that they regulate the general splicing process to adapt to different cell cycle phases or modulate periodic alternative splicing along cell cycle, which is consistent with previous studies[42,43].

Besides RBPs, N6-methyladenosine (m6A) modification, one of the most abundant modifications on mammalian mRNA, has also been shown to regulate the fate of RNA by recruiting different readers[44–46]. One of the well-known readers of m6A modification is *YTHDF2*, and it has been shown to affect the RNA stability[47,48]. Here, we observed that *YTHDF2* target genes[48] are significantly enriched in the genes with m6A modification[49,50] in Modules 3, 4, and 5 (Supplementary Fig. 15c), and meanwhile, the gene expression of *YTHDF2* is also cell cycle dependent (Supplementary Fig. 15d). These results showed that besides RBPs, RNA modification is also associated with the regulation of Type II CCGs. Overall, our analysis suggested that RNA binding proteins and m6A modification of RNA play important roles in regulating the expression of Type II CCGs along the cell cycle.

### RNA velocity analysis of oncogene-induced senescence

To further test the ability to dissect the gene expression kinetics and gene regulation underlying cell-state transition, we applied snapTotal-seq to characterize the gene regulation in the entry into the oncogene-induced senescence (OIS). Here, we utilized the HPNE cells with inducible KRAS[G12D] expression[51]. After the activation of KRAS[G12D], we collected the cells on Days 1, 2, 3, and 5 (Fig. 5a), and in total, we sequenced 642 cells using snapTotal-seq. The cells were plotted on the UMAP and were labeled by the time points (Fig. 5b) and the assigned cell cycle stages (Fig. 5c). Interestingly, we observed that the cells are distributed in two clusters. One cluster (top right cluster in Fig. 5c) is mainly composed of G1, S, and G2/M phases, while the other one (bottom-left cluster in Fig. 5c) is mainly composed of the G0 population based on reCAT analysis. By comparing the percentage of different cell cycle stages at each time point, we observed a gradual increase of the G0 cells from days 1 to 5 (Supplementary Fig. 16a), suggesting

that the cells continuously entered the G0 phase following the induction of oncogenic KRAS.

Next, we applied RNA velocity analysis to construct the trajectory of OIS. We observed a clear trajectory within the G0 population indicated by the bold arrow in Fig. 5d. The trajectory follows the time order of days 1, 2, 3, and 5 (Fig. 5d and Supplementary Fig. 16b). Therefore, the bottom-left region of the G0 cluster (dashed circle in Fig. 5d) corresponds to the endpoint: the senescent state. The successful trajectory inference was confirmed by the velocity confidence scores (0.88 ± 0.050) (Supplementary Fig. 16c). The inferred latent time also agreed with the experimental time points (Supplementary Fig. 16d).

### Comparative analysis between unspliced and spliced RNAs along OIS

Next, to investigate the gene expression dynamics during OIS, we performed the differential gene expression analysis for both spliced RNA and unspliced RNA based on the trajectory within the G0 population established by RNA velocity analysis (Fig. 5e and Supplementary Data 6). Similar to cell cycle analysis, we define the genes with significant changes in both spliced RNA and unspliced RNA as Type I DEGs, the genes with only significant changes in spliced RNA as Type II DEGs, and the genes with only significant changes in unspliced RNA as Type III DEGs. In total, we identified 2113 genes as Type I DEGs, 2738 genes as Type II DEGs, and 1087 genes as Type III DEGs (Fig. 5e).

### Identification of five gene expression kinetic modules in OIS

We focused on the transcriptional regulation of the Type I DEGs during OIS. Through an analysis of their transcriptional dynamics, we first detected five kinetic modules among Type I DEGs (Fig. 5f, g and Supplementary Data 7). We noticed that Modules 1 and 2 were significantly downregulated along the senescence entry, and they were significantly enriched with the genes involved in cell adhesion and extracellular matrix organization (Fig. 5f, g and Supplementary Fig. 16e). Conversely, the expression level of Module 3 was transiently downregulated at the early stage, followed by a rapid rebound to its original level (Fig. 5f and Supplementary Fig. 16f). Lastly, the expression of Modules 4 and 5 demonstrated significant upregulation along the trajectory, underscoring their pivotal roles in establishing OIS (Fig. 5f and Supplementary Fig. 16g). Specifically, Module 4 that is enriched with the genes involved in the immune response (Supplementary Fig. 16g), was quickly activated after the induction of oncogenic stress, followed by a linear increase in their expression levels. In contrast, the proteotoxic stress response pathway (i.e., Module 5, Supplementary Fig. 16g) exhibited an initial slow activation, which was succeeded by a rapid elevation in expression levels at the middle time point.

### Identification of TF Hubs underlying the kinetic modules by LASSO regression

To identify the TF hubs that regulate these kinetic modules, we established the association between TFs and their target genes based on the LASSO analysis, the same as analyzed in the cell cycle (Fig. 3a). In total, we identified 112 TFs whose linked genes were significantly enriched with Type I kinetic modules (FDR < 0.01). Among them, 88 out of 112 TF modules and 4428 out of 5413 TF-gene links were also captured by GENIE3 (Supplementary Fig. 17), suggesting the high confidence of the TF modules identified here. These TFs were then grouped into 3 distinct regulatory hubs based on their activities along the trajectory of OIS (Fig. 5h–I and Supplementary Data 8). We observed that Hub 1 is mainly associated with kinetic Modules 1 and 2, Hub 2 is mainly associated with kinetic Module 3, and Hub 3 is mainly associated with kinetic Modules 4 and 5.

Next, we identify TF–TF associations within the TF hubs or between the TF hubs using LASSO analysis. In Fig. 5j, we colored

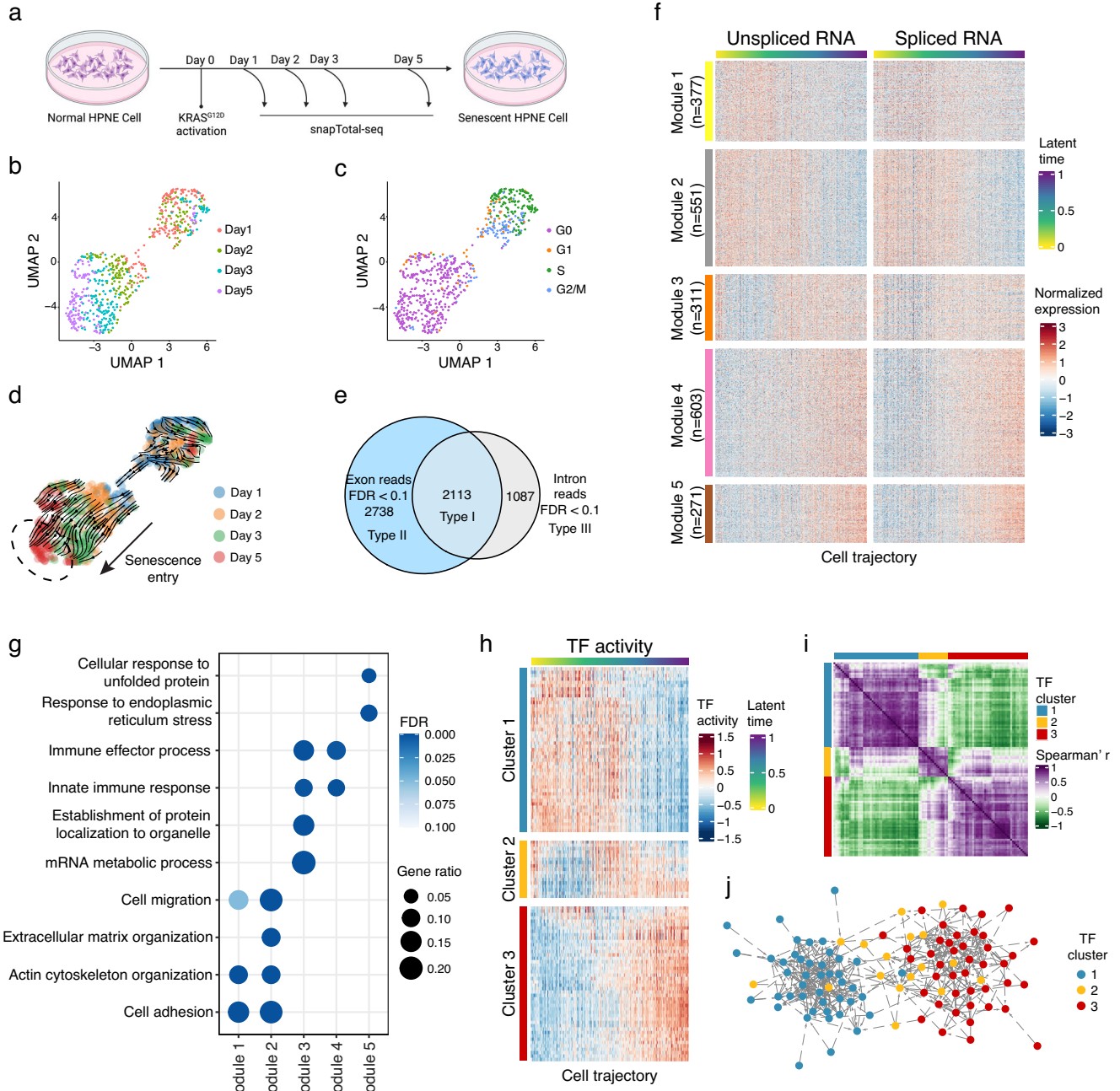

**Fig. 5 | Transcriptional dynamics during oncogene-induced senescence.**
**a** Experimental scheme for studying gene expression dynamics of oncogene-induced senescence. The plot was created with BioRender.com. UMAP visualization of the HPNE cells with KRAS[G12D] overexpression. The cells are colored by the time points (**b**) or the cell cycle stages (**c**). **d** The projected velocity trajectory of oncogene-induced senescence by RNA velocity analysis. The cells are colored by the time points. **e** Differential gene expression analyses identified the genes with significant changes at both spliced RNA and unspliced RNA levels (Type I), only at the spliced RNA level (Type II), and only at the unspliced RNA level (Type III) along

oncogene-induced senescence. **f** The gene expression heatmap of the genes with significant changes at both spliced RNA and unspliced RNA levels (Type I DEGs) along the latent time. **g** The GO enrichment in five kinetic modules of Type I DEGs. **h** Heatmap of the activities of all enriched TFs along cell trajectory. **i** Heatmap of the correlation coefficients between the activities of each pair of TFs. **j** TF association network established based on the TF–TF links identified with LASSO regression. **a** Created with BioRender.com released under a Creative Commons Attribution-NonCommercial-NoDerivs 4.0 International license.

TFs based on their hubs as determined above. As a result, we observed that TFs from the same hub tend to form densely inter-connected sub-networks (Student's t-test, $p < 2.2e - 16$ for all TF hubs). More interestingly, we observed that Hub 2 is located between Hubs 1 and 3 (Fig. 5j), indicating their important roles in the shifting process from one regulatory controller to another regulatory controller.

## Validation of specific regulations based on ChIP-seq data and motif analysis

To further pin down the regulatory relationships between the TFs, we examined whether associated genes were significantly enriched with either the ChIP-seq validated binding targets or the gene promoters carrying the corresponding binding motif. As a result, we found 25 TFs with significant enrichment (Supplementary Data 9). We also labeled

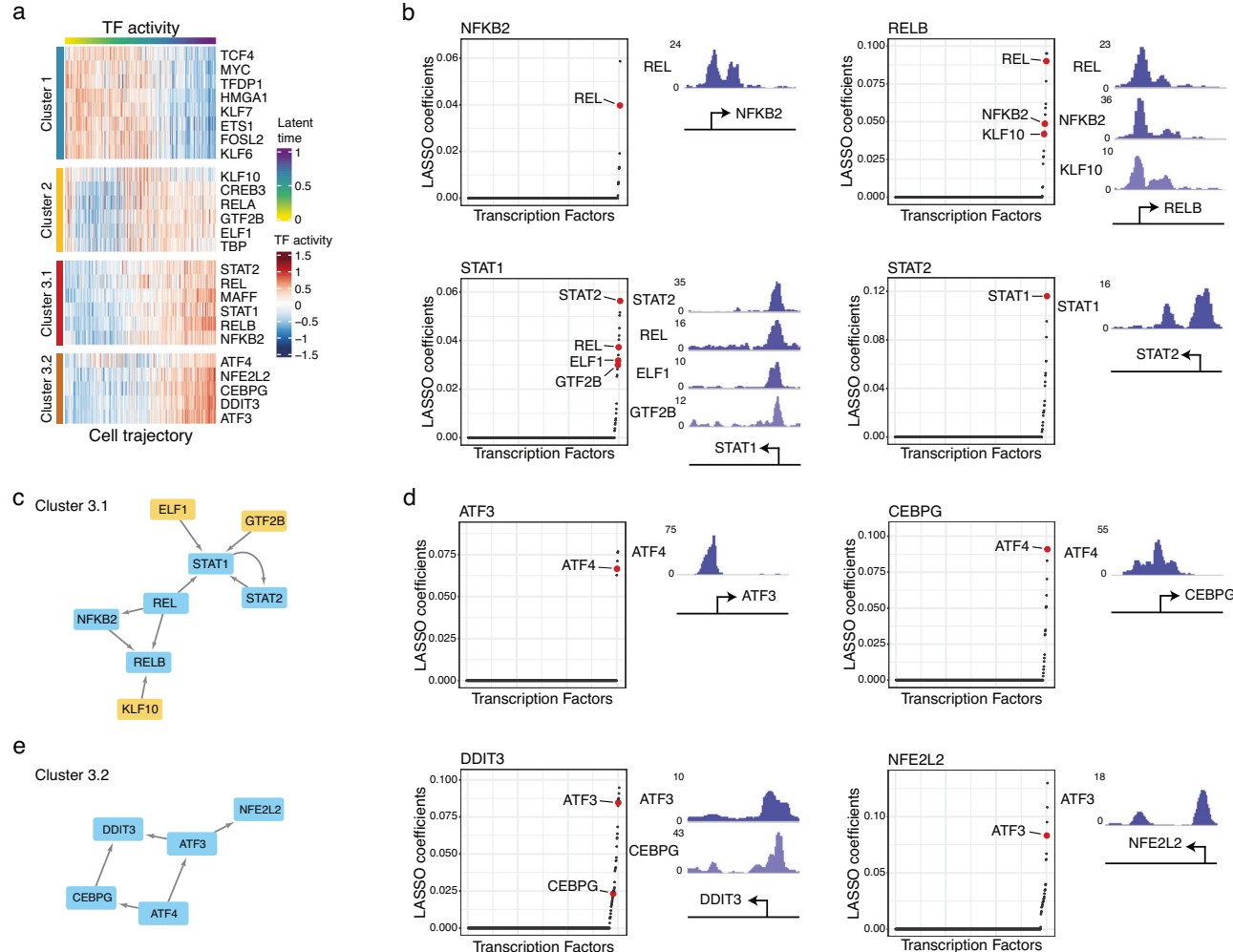

**Fig. 6 | Identifying key TF regulatory networks driving OIS. a** Heatmap of the activities of the verified TFs. **b** Establish the regulatory links between the TFs within TF Hub 3.1 by correlating the changes in the unspliced RNA of the TF of interest with the changes in the spliced RNA of the other TFs. The direct regulatory links (colored in red) were identified by ChIP-seq verification. The corresponding ChIP-seq peaks were plotted on the right. **c** The TF regulatory network of TF Hub 3.1. The regulatory network was established based on the direct regulatory links between different TFs.

The TFs belonging to TF Hub 2 are colored in yellow. **d** Establish the regulatory links between the TFs within TF Hub 3.2 by correlating the changes in the unspliced RNA of the TF of interest with the changes in the spliced RNA of the other TFs. The direct regulatory links (colored in red) were identified by ChIP-seq verification. The corresponding ChIP-seq peaks were plotted on the right. **e** The TF regulatory network of TF Hub 3.2. The regulatory network was established based on the direct regulatory links between different TFs.

these 25 verified TFs based on their respective TF hubs described above (Fig. 6a).

Notably, based on the gene expression kinetics of the 25 TFs, Hub 3 can be further divided into two sub-hubs (Hub 3.1 and Hub 3.2). Intriguingly, Hub 3.1 predominantly consisted of TFs regulating the immune response pathway, while Hub 3.2 was comprised of TFs linked to the integrated stress response pathway (Fig. 6a). This observation highlights the distinct and specific functional roles of different TF hubs during the entry into the OIS.

In Hub 3.1, we identified *REL* as a critical player that orchestrates the activities of both the NF-kB pathway and the STAT1/STAT2 pathway (Fig. 6b, c). Interestingly, we also observed that the TFs belonging to Hub 2 (colored in yellow in Fig. 6c) were also involved in the regulation of *RELB* and *STAT1* through the direct binding to their promoter regions, confirming the previous observation of the regulatory connection between Hubs 2 and 3.

In Hub 3.2, *ATF4* was identified as the most upstream regulator within this regulatory network. Our data revealed that *ATF4* directly regulated the expression of *CEBPG* and *ATF3* (Fig. 6d, e), both of which play critical roles in regulating cellular stress response[52,53]. The

activation of *CEBPG* and *ATF3*, in turn, led to the upregulation of *DDIT3*, which is involved in the regulation of ER stress response[54,55], and *NFE2L2*, which regulates the oxidative stress response[56] (Fig. 6d, e). This cascade of regulatory events underscores *ATF4*'s central role in orchestrating the stress response pathway through the regulation of key downstream genes, aligning with previous findings[57–60]. Overall, our data unveiled the core TF network that governs the entry into cellular senescence after the induction of oncogenic stress.

## RNA binding proteins (RBP) and alternative polyadenylation (APA) based post-transcriptional regulations during OIS

As shown in Fig. 5e, a substantial proportion of genes were Type II DEGs, whose gene expression is dominantly affected by post-transcriptional regulation. Based on the dynamic changes in their spliced RNA abundance, we classified these genes into five kinetic modules with significant enrichment of different pathways (Supplementary Fig. 18 and Supplementary Data 7).

Next, we examined the potential mechanisms driving the differential gene expression of Type II DEGs. We first compared the binding targets of the known 150 RBPs with these 5 kinetic modules, and we

identified the significant enrichment of the binding targets of certain RBPs in Modules 1, 2, and 3 (Supplementary Fig. 19a). Interestingly, a notable proportion of the RBPs enriched in Modules 1 and 2 are splicing regulators[36] (Supplementary Fig. 19b), indicating that the splicing of the target genes detected in these two modules was regulated during oncogene-induced senescence.

Besides the regulation of RNA fate by RBPs, alternative polyadenylation (APA) has also been reported as a common mechanism to regulate gene expression in response to cellular stress[42]. The lengthening of 3′UTR associated with the usage of distal polyA sites could increase the binding sites for RBPs or miRNAs and hence promote RNA degradation[61–63]. Interestingly, Chen et al. have reported a global lengthening of 3′UTR in replicative senescence[64]. To test whether APA is also involved in the post-transcriptional regulation in OIS, we evenly separated the cells into five time windows (intervals) along the trajectory and used DaPars2[65] to derive the preference of polyA site usage within each interval. Indeed, a substantial proportion of the genes were found to use multiple polyA sites (Supplementary Fig. 20a), which makes them qualified for APA analysis.

As a result, we observed different trends in alternative polyadenylation among the five kinetic modules. In Module 1, we observed an increase in the usage of distal polyA sites in intervals 2 and 5, which corresponds to the decrease in gene expression at the corresponding intervals (Supplementary Fig. 20b, c). In Modules 3, 4, and 5, we observed a clear decrease in the usage of distal polyA sites at the late intervals (Supplementary Fig. 20g–i), which is consistent with the significant increase in the expression levels of these genes at the late time points (Supplementary Fig. 20f). It is worth noting that in Module 2, we only observed slight changes in the usage of polyA sites along the trajectory (Supplementary Fig. 20d, e), suggesting that the expression of these genes was not mainly regulated by APA. Overall, these observations confirm APA as a general post-transcriptional regulation during oncogene-induced senescence and it can be captured by snapTotal-seq.

## Discussion
In summary, we developed a high-sensitivity single-cell total-RNA-seq method (snapTotal-seq) by combining the multiple annealing chemistry and template-switching chemistry into a one-step reaction. Benchmark analysis showed that our method, with a simplified chemistry, achieves a gene detection sensitivity similar to VASA-seq, while significantly surpassing the widely used oligo-dT-based methods, particularly for moderately and lowly expressed genes.

Furthermore, by taking the intrinsic cell cycle as an example, we conducted a comprehensive analysis to compare both oligo-dT-based and total-RNA-based scRNA-seq methods for their performance in characterizing gene expression dynamics through RNA velocity analysis. As a result, with the efficient detection of both unspliced RNAs and spliced RNAs, snapTotal-seq demonstrated significant advantage in reconstructing the underlying dynamic process of cell cycle progression, in comparison to oligo-dT-based methods.

With the established cell trajectory by RNA velocity, we successfully identified a large number of noncanonical cell cycle genes and unveiled the related biological processes intricated with cell cycle progression. It is also worth noting that a small fraction of the canonical cell cycle genes were not captured in our data. First, the genes regulated at the protein level throughput the cell cycle were not detected by our analysis. For instance, cyclin D and CDK4/6, which drive G1/S transition, undergo tight regulation by either ubiquitin-proteasomal pathway[66,67] or inhibitory protein binding[68–70]. Consistently, no significant changes at the RNA level were detected in our data for these genes. Second, previous studies have shown that some regulators of cell cycle are cell-type specific[69,71,72]. Since the list of canonical cell cycle gene list is curated from previous studies based on

a variety of biological contexts, it could include cell-type specific cell cycle regulators which may not participate in the cell cycle regulation in HEK293T cell line we characterized here.

Beyond the identification of differentially expressed genes, we were able to directly link the expression of TFs to the transcriptional activities of their target genes. Here we performed LASSO regression of the transcriptional dynamics of a specific TF against the expression of other TFs. As a result, we demonstrated the ability of snapTotal-seq to derive the inference hubs of TFs and identify the key TFs that control the cell-state transition. In contrast, with methods that only profile the spliced RNAs, this type of analysis is unfeasible.

Besides the regression-based approach used here, non-parametric algorithms have been developed to infer gene regulatory network (GRN) through gene expression data[30,73–75]. In comparison to regression-based approaches, these non-parametric methods aim to capture more intricate dependencies between different genes, including non-linear ones. To evaluate the performance of different approaches on identifying TF-gene regulation from our data, we employed GENIE3[22,30], a non-parametric method, to compare with the regression-based method. Overall, most of high confidence TF-gene links were captured by both methods, suggesting the robustness of GRN inference on our data. Consistent with the features of non-parametric approaches, GENIE3 identified more TF-gene links compared to LASSO regression, potentially representing non-linear dependencies due to overall lower scores. Although further analyses were not conducted here, future investigations on these non-linear dependencies are greatly desired to unveil the intricacies of gene regulation.

To further illustrate the performance of snapTotal-seq, we applied it to the process of oncogene-induced senescence. And we successfully pinpointed the key TF regulatory hubs that drove the cells into the senescent state. In summary, our work demonstrates the versatility and potential of snapTotal-seq in advancing our understanding of gene expression dynamics and, more importantly, the regulatory hubs underlying cell-state transition.

## Methods
### Cell cultures
HEK293T (CRL-3216, obtained from ATCC) cells were cultured in DMEM with 10% fetal bovine serum (FBS, Life Technologies) and were passaged every 2 days with 0.05% trypsin (Corning®). hTERT-HPNE (CRL-4023, obtained from ATCC) cell line was cultured in the medium with 75% DMEM without glucose (Sigma), 25% Medium M3 Base (INCELL Corp.), 2 mM L-glutamine (Sigma), 1.5 g/L sodium bicarbonate (Sigma), 5% FBS (Life Technologies), 10 ng/mL human recombinant EGF (ThermoFisher), 5.5 mM D-glucose (Sigma) and 750 ng/mL puromycin (InvivoGen), as recommended by ATCC. The cells were passaged every 2–3 days with 0.25% trypsin (Corning®).

The induction of oncogene-induced senescence was performed as described in previous publicaiton[51]. Briefly, wild-type HPNE cells with inducible KRAS$^{G12D}$ (iKRAS-HPNE) were cultured with doxycycline (6 μg/mL) to activate the expression of KRAS$^{G12D}$. The cells were collected on days 1, 2, 3, and 5 after the induction of KRAS$^{G12D}$ expression for single-cell RNA-seq experiments.

U2OS (HTB-96, obtained from ATCC) cell line was cultured in McCoy's 5a medium (ATCC) supplemented with 10% FBS (Life Technologies), and the cells were passaged every 2–3 days with 0.25% trypsin (Corning®). NIH/3T3 (CRL-1658, obtained from ATCC) cell line was cultured in DMEM with 10% fetal bovine serum (FBS, Life Technologies) and was passed every 2 days with 0.25% trypsin (Corning®). To perform single-cell RNA-seq experiments, the cells were trypsinized and resuspended in PBS (Corning®). The cells were then sorted into the 96-well plates with 1 μL lysis buffer per well by using BD Aria II with a 130 μm nozzle.

## Cell lysis, reverse transcription, and amplification by snapTotal-seq

The cell lysis buffer consisted of 0.7 μL of 1.8% Triton-X (Sigma), 0.025 μL of 0.1 M DTT (Invitrogen), 1 U RNaseOUT (Invitrogen), 0.05 μL of dNTP (10 mM each) and 0.2 μL of primer mix (1.5 μM of GTG AGT GAT GGT TGA GGA TGT GTG GAG N5 T12, 5 μM of GTG AGT GAT GGT TGA GGA TGT GTG GAG N5 T3, 5 μM of GTG AGT GAT GGT TGA GGA TGT GTG GAG N5 G3). 10 μL of mineral oil (Sigma) was added to prevent the evaporation in the following steps. The lysis was performed at 42 °C for 3.5 min. After lysis, the plate was immediately placed on the ice for 1 min. Reverse transcription mix which contained 0.4 μL of 5X M-MLV reverse transcriptase buffer (250 mM Tris-HCl, 375 mM KCl, 15 mM $MgCl_2$, Invitrogen), 0.1 μL of 0.1 M DTT (Invitrogen), 2 U RNaseOUT (Invitrogen), 10 U Maxima H Minus Reverse Transcriptase (Thermo Scientific) and 0.4 μL of 0.1% Triton-X (Sigma), was then added to each well. The reverse transcription and template-switching step was carried out with 10 cycles of 8 °C for 12 s, 15 °C for 45 s, 20 °C for 45 s, 30 °C for 30 s, 42 °C for 2 min, and 50 °C for 3 min, followed by 50 °C for 15 min. The reverse transcriptase was inactivated at by incubating at 74 °C for 25 min.

After that, PCR amplification mix, which consisted of 10 μL of 5X GoTaq Flexi buffer, 4 μL of 25 mM $MgCl_2$, 1 μL of dNTP (10 mM each), 0.25 μL of 100 μM GAT primer (GTG AGT GAT GGT TGA GGA TGT GTG GAG), 1.75 U GoTaq Flexi DNA polymerase, 2.5 μL of 20X EvaGreen Dye (Biotium) and 29.9 μL of RNase-free $H_2O$, was added to each well. The amplification was carried out on a Real-time PCR machine. The PCR program was as follows: 95 °C for 2 min, 23–26 cycles of 95 °C for 20 s, 63 °C for 20 s and 72 °C for 2 min, 72 °C for 5 min.

Purification was carried out with 1.2X Ampure XP beads (Beckman Coulter). The samples were mixed with Ampure XP beads and incubated for 10 min at room temperature. Then, the plate was placed on a 96-well magnetic stand, and the supernatant was removed. To remove the residual mineral oil, we washed the beads with 2-propanol (Sigma) twice, followed by the wash with 100% ethanol (Koptec). Next, the beads were washed twice with 80% ethanol. Finally, the amplified products were eluted in 25 μL of RNase-free $H_2O$.

Next, the amplified products from each cell were tagged with cell-specific barcodes by double-strand conversion (DSC). Specifically, for each cell, 10 μL of amplified product was mixed with 2 μL 10X ThermoPol Buffer, 0.4 μL of dNTP (10 mM each), 1 μL of 10 μM DSC primer with cell barcode (GTGTGCTCTTCCGATCT NNNNNNNN AGGAG AGT GTG AGT GAT GGT TGA GGA TGT GTG GAG), 0.4 U Deep Vent (exo-) DNA polymerase (New England BioLabs) and RNase-free $H_2O$ to 20 μL. The DSC program was as follows: 95 °C for 1 min, 15 cycles of 63 °C for 25 s and 72 °C for 1 min, 72 °C for 3 min. 1 μL of 0.2 M EDTA (Sigma) was then added to each cell to stop the reaction.

Here, we would like to point out the design principle for adding cell barcode sequence after the PCR amplification. Since the adapter sequences of our cDNA amplicon products have complementary on both ends, if we introduce the cell barcode during the PCR amplification step, it will result in the cell barcode being added to both ends of the amplicon products. As a result, it would create a substantial stem loop structure that essentially hinders the efficient amplification. In contrast, when we introduce cell barcode via the extra double-strand conversion step after the PCR amplification, the cell barcode is only be added to one end of the amplicon.

## Sequencing library construction

The cells with different cell barcodes were pooled together (4 μL of reaction product per cell) and were purified with 1.1X Ampure XP beads. For each pooled library, ~350 ng of purified DNA products were then sonicated to 150–250 bp (Covaris S220). The sonicated DNA was purified with 1.8X Ampure XP beads. Following that, another step of DSC was performed to enrich the DNA fragments with UMI and cell barcode. Briefly, the sonicated product was mixed with 3 μL of 10X ThermoPol Buffer, 0.6 μL of dNTP (10 mM each), 1.5 μL of 10 μM primer (GCACGACATCTGCTAACGCAGTA GTGTGCTCTTCCGATCT), 0.6 U Deep Vent (exo-) DNA polymerase and $H_2O$ to 30 μL. The reaction was carried out as follows: 95 °C for 1 min, 6 cycles of 56 °C 25 s and 72 °C 30 s, 72 °C for 3 min. The products were then purified with 1.4X Ampure XP beads. The purified products were subjected to dA tailing with 2 μL of 10X NEBuffer 2 (New England BioLabs), 0.1 μL of 100 mM dATP, and 2.5 U Klenow Fragment (3' to >5' exo-) (New England BioLabs) and $H_2O$ to 20 μL, by incubating at room temperature for 30 min.

After purifying the products with 1.4X Ampure XP beads, we performed the ligation at room temperature for 20 min. The ligation reaction mix included 13 μL of 2X Quick Ligase reaction buffer, 0.5 μL of 50 mM Y-shape adapter, 0.5 μL of Quick Ligase (New England BioLabs), and 12 μL of dA-tailing product. The ligation reaction was quenched by adding 5 μL of 0.2 M EDTA (Sigma) and was purified by 1.2X Ampure XP beads. The amplification of the ligation products was then performed with the program as follows: 95 °C for 2 min, 10 cycles of 95 °C 20 s, 61 °C 20 s and 72 °C for 1 min, and 72 °C for 3 min for final extension. The amplification mix consisted of 5 μL of ThermoPol Buffer, 1 μL of dNTP (10 mM each), 2 μL of 10 μM forward primers (GCA CGA CAT CTG CTA ACG CAG TA), 2 μL of 10 μM reverse primers (AAT GAT ACG GCG ACC ACC GAG A), 1 U of Deep Vent (exo-) DNA polymerase, 33.5 μL of $H_2O$ and 6 μL of ligation products.

After the amplification products were purified with 1.2X Ampure XP beads, duplex-specific nuclease (DSN) treatment was applied to remove ribosomal reads. Specifically, 100 ng of amplified products was mixed with 2 μL of 10X DSN buffer (Evrogen) and $H_2O$ to 20 μL. The DNA was denatured at 95 °C for 30 s, followed by incubation at 80 °C for 3 h. After that, 1 μL of preheated duplex-specific nuclease (Evrogen) was added to the reaction and incubated at 80 °C for 15 min. To quench the reaction, 4 μL of 0.2 M EDTA was added at 80 °C, and the reaction was then put on ice immediately. The products were purified with 1.2X Ampure XP beads.

Following that, an enrichment PCR was carried out to enrich the undigested DNA fragments. The reaction mix consisted of 2.5 μL of 10X ThermoPol Buffer, 0.5 μL of dNTP (10 mM each), 0.75 μL of 10 μM forward primers (GCA CGA CAT CTG CTA ACG CAG TAG TGT GCT CTT CCG ATC T), 0.75 μL of 10 μM reverse primer (AAT GAT ACG GCG ACC ACC GAG A), 0.5 U Deep Vent (exo-) DNA polymerase, 12.25 μL of $H_2O$ and 8 μL of DSN treatment product. The program was as follows: 95 °C for 2 min, 5 cycles of 95 °C for 20 s, 63 °C for 20 s and 72 °C 1 min, 72 °C 3 min for final extension. The amplified products were purified with 1.2X Ampure XP beads.

A final PCR step, which consisted of 2.5 μL of 10X ThermoPol Buffer, 0.5 μL of dNTP (10 mM each), 0.75 μL of 10 μM forward primers (CAA GCA GAA GAC GGC ATA CGA GAT GCA CGA CAT CTG CTA ACG CAG TA), 0.75 μL of 10 μM reverse primers (AAT GAT ACG GCG ACC ACC GAG A), 0.5 U Deep Vent (exo-) DNA polymerase, 19.25 μL of $H_2O$ and 1 μL of purified DNA product, was performed to add the sequencing adapter. The program runs as follows: 95 °C for 2 min, 5 cycles of 95 °C for 20 s, 61 °C for 20 s and 72 °C 1 min, 72 °C 3 min for final extension.

The libraries were sequenced on NextSeq 500 machine with customized sequencing primers as follows: Read 1 sequencing primer: ACA CTC TTT CCC TAC ACG ACG CTC TTC CGA TCT (the same as Illumina Tru-seq i5 sequencing primer), Read 2 sequencing primer: AGA GGT GAG TGA GTG ATG GTT GAG GAT GTG TGG AG, Index i5 sequencing primer: AGA TCG GAA GAG CGT CGT GTA GGG AAA GAG TGT, Index i7 sequencing primer: CTC CAC ACA TCC TCA ACC ATC ACT CAC TCA CCT CT. Read 1 sequenced the captured RNA sequence, while Read 2 sequenced the UMI.

Alternatively, the snapTotal-seq libraries can be constructed into Illumina standard libraries by replacing the forward primer in the final PCR step with Truseq-P7 primer (CAAGCAGAAGACGGCATACGAG

ATCGAGTAATGTGACTGGAGTTCAGACGTGTGCTCTTCCGATCT). The PCR program runs as follows: 95 °C for 2 min, 5 cycles of 95 °C for 20 s, 55 °C for 20 s and 72 °C 1 min, 72 °C 3 min for final extension. The libraries can then be sequenced with Illumina standard sequencing primers. Read 1 sequenced the captured RNA sequence, while Read 2 sequenced the cell barcode (bases 1–8) and UMI (bases 44–48). The sequence of DNA oligos used in snapTotal-seq is provided in Supplementary Data 12.

### Reads alignment and gene expression calculation
Read 1 was mapped to human genome assembly (GRCh37) by using STAR (v2.5.3a)[76]. The uniquely mapped reads were then mapped to the gene annotations of GENCODE (v19) by using htseq-count[77] with "intersection-strict" mode and with option "--stranded = no". To discern the amplicons from exons and introns, the "transcript" feature and the "exon" feature were used respectively. The UMI sequences (the first five bases of Read 2) of the reads mapped to the gene regions (either the "transcript" feature or the "exon" feature) were extracted. The reads were then grouped by the UMI sequence and the gene that they were mapped to, as these reads were derived from the same original cDNA amplicon. If all the reads within the group were mapped to the exon regions of the corresponding gene, the original amplicon was classified as an exonic amplicon. Otherwise, the original amplicon was classified as an intronic amplicon. The number of exonic amplicons and intronic amplicons were then counted for each gene, and the exonic UMI count matrix and intronic UMI count matrix were generated.

### Normalization, PCA, cell cycle analysis, and RNA velocity analysis
The genes that were detected with more than one UMI in at least five cells in exon data were kept. The rest of the genes were defined as lowly expressed genes and were discarded in the following analysis. The mitochondria genes were also removed before the normalization step. To normalize the gene expression data across different cells, we divided the count of UMIs of each gene by the total UMIs detected in each cell and multiplied by the average UMI number of all cells. The normalization step was performed on exonic data and intronic data separately. PCA was performed on exon-based gene expression data by using Seurat package[78]. Briefly, the exonic normalized gene expression data were log-transformed and scaled. The top 500 most variable genes were selected by using "FindVariableFeatures" function to perform PCA. Elbow plot was then used to determine the dimensionality of the dataset. The percentage of variance explained by PC1 and PC2 was then calculated. UMAP was performed for dimensional reduction. Unsupervised clustering was performed by constructing the KNN graph ("FindNeighbors" function) with k.param = 10 and then clustering with Louvain algorithm ("FindClusters" function) with resolution = 0.4. The cell cycle analysis was performed with reCAT[17] by using the exon-based gene expression data. The log-transformed normalized gene expression was used as the input. RNA velocity analysis was performed by using scVelo package[12]. The raw count matrices were used as the input. To filter the lowly expressed genes, the "min_shared_counts" was set as 30 and the "min_cells_u" as 5. After normalization and log-transformation, the velocities were projected by using the top 2000 most variable genes and the "dynamical modeling" mode with the following parameters: n_pcs = 2, n_neighbors = 20, fit_basal_transcription = True. Latent time was calculated based on the projected RNA velocities.

### Cell cycle trajectory-based differential gene expression analysis
The differential gene expression analysis was performed on exonic data and intronic data, respectively, by using tradeSeq package[18]. The exonic or intronic raw count matrix was used as the input. The pseudotime of each cell was derived from the cell cycle trajectory inferred by RNA velocity. The parameter "nknots" in function "fitGAM" was set as 5. The "associationTest" function was used to identify the genes which were differentially expressed along the pseudotime. The genes whose exonic normalized gene expression values and intronic normalized gene expression values were ≥3 in at least 10 cells were kept. To perform the multiple test correction, the false discovery rate (FDR) was calculated by using p.adjust function in R. The differentially expressed genes (DEG) were identified using FDR < 0.1 to facilitate the detection of the genes with moderate changes during cell cycle. The DEGs identified with a more stringent criteria (FDR < 0.05) are examined in Supplementary Note 3. To verify the results of DEG analysis, we calculated the fold changes in gene expression along the cell cycle for all genes. To account for the variations in single-cell data, we evenly separated the cells into six intervals along the cell cycle, and the average expression levels were calculated for each interval. The fold changes between the highest expression level and the lowest expression level among these intervals were then calculated. The lowly expressed genes with maximum expression lower than 1 were excluded.

### Analysis of Smart-seq3, CEL-Seq2, VASA-seq, and Smart-seq-total
To perform the benchmark analysis on Smart-seq3, we reanalyzed the published HEK293FT dataset generated by Smart-seq3. The raw fastq files were first demultiplexed based on the cell indexes. The UMI reads were identified based on the adapter sequence. The adapter sequence and the UMI sequence were first trimmed by using seqtk. The trimmed reads were then mapped to human genome assembly (GRCh37) by using STAR (v2.5.3a). Following that, the uniquely mapped reads were assigned to exon or intron regions by using htseq-count as described above. The reads were collapsed if the hamming distance of UMIs ≤ 1, and the exonic UMI count matrix and intronic UMI count matrix were generated. Two potential outliers that were identified based on the PCA plot in the initial analysis were discarded. The normalization, PCA, cell cycle analysis, and RNA velocity analysis were then performed as described above.

To perform the benchmark analysis on CEL-Seq2, we reanalyzed the published HEK293 dataset generated by CEL-Seq2. To avoid potential batch effects, only the cells collected in the mixture 2 experiment, as described in the original study, were used. The demultiplexed reads were mapped to human genome assembly (GRCh37) by using STAR (v2.5.3a). Next, the uniquely mapped reads were assigned to exons or introns, as described above. The reads that were mapped to the same gene were collapsed based on their UMI sequences, which generated the exonic UMI count matrix and intronic UMI count matrix. The potential outliers that were identified based on the PCA plot in the initial analysis were discarded. The normalization, PCA, cell cycle analysis, and RNA velocity analysis were then performed as described above.

To reanalyze the published HEK293T dataset generated by VASA-seq (plate version), the raw fastq files were first demultiplexed based on the cell indexes. The demultiplexed reads were mapped to human genome assembly (GRCh37) by using STAR (v2.5.3a). Next, the uniquely mapped reads were assigned to exons or introns, as described above. The reads that were mapped to the same gene were collapsed based on their UMI sequences, which generated the exonic UMI count matrix and intronic UMI count matrix. The cells with low sequencing depth (total exon UMI < 7500) or high genomic DNA contamination were discarded. The potential outliers which were identified based on the initial PCA were also discarded. The normalization, PCA, cell cycle analysis, and RNA velocity analysis were then performed as described above.

To analyze the published HEK293T dataset generated by Smart-seq-total, we first trimmed the poly(A) sequences from the raw reads using Cutadapt (v3.4)[79]. The trimmed reads were then mapped to

human genome assembly (GRCh37) by using STAR (v2.5.3a). The uniquely mapped reads were assigned to exon or intron regions by using htseq-count as described above.

## Benchmark analysis on gene detection sensitivity

Gene annotation was obtained from GENCODE (v19). To evaluate the gene detection of different RNA biotypes, we classified all genes into protein coding, non-coding (including lincRNA, pseudogene and antisense genes), small RNA (including snRNA, snoRNA and miRNA), and other types. To evaluate the detection of genes with different expression levels, for each dataset, we discarded the genes expressed in <20% of cells and grouped the rest of genes based on their average expression levels. All the datasets were normalized using a common scaling factor of 100,000 UMIs, to enable the application of a common cutoff for gene classification. Detection rates for each groups of genes were then calculated. To evaluate the detection bias towards RNA length, genes were evenly categorized into three categories based on their length. In exon-read analysis, only the length of mature RNA (i.e., the sum of all exons) was considered. In intron-read analysis, the entire gene length was considered. Read distribution analysis was carried out using RseQC[80] package with hg19 RefSeq gene model.

## Validation of the identified cell cycle dependent genes by qRT-PCR

HEK293T-FUCCI cell line was established by using the FastFUCCI plasmid (addgene #86849). The cells with successful transduction were selected by culturing the cells with puromycin (1 μg/mL) for 4 days. After collecting the cells of different cell cycle phases by FACS, RNA was extracted by using TRIzol reagent (Invitrogen) according to the manufacturer's instructions. The reverse transcription reaction and qPCR were then carried out by using iScript™ reverse transcription supermix (Bio-Rad) and iTaq universal SYBR green supermix (Bio-Rad), respectively. The qPCR program was as follows: 94 °C for 2 min, 40 cycles of 94 °C for 20 s, 58 °C for 20 s, and 72 °C for 20 s. *ATP5F1* was used as the internal control, as its expression remained unchanged along the cell cycle. The primer sequences used for qRT-PCR are provided in Supplementary Data 10.

## Calculation of cross-correlation

To quantify the coupling between the gene expression changes in unspliced RNA and spliced RNA, we borrowed the metrics of "cross-correlation" from the field of signal processing, which is defined as a measure of similarity between two signals. For each gene module, the average z-score of gene expression along the cell cycle trajectory was calculated for unspliced RNA and spliced RNA, respectively. Following that, the cross-correlation between the expression curves of unspliced RNA and spliced RNA was calculated by using numpy.correlate function in Python.

## Functional analysis on cell cycle genes (CCGs)

The list of canonical CCGs was obtained by combining the gene list of the cell cycle pathway in the Gene Ontology database[81], the gene list of the cell cycle pathway in the Reactome database[82] and the genes reported in Cyclebase[83]. The Gene Ontology enrichment analysis was performed by using "Hypergeometric" model in "goseq" R package[84]. The cellular localization was obtained from the cellular compartment category of the GO database. Fisher's exact test was used to determine the differential enrichment of canonical CCGs and noncanonical CCGs in different cellular compartments. The "endomembrane system" consisted of "GO:0031226", "GO:0005887", "GO:0031224", "GO:0016021", "GO:0000139", "GO:0098791", "GO:0098588", "GO:0005783", "GO:0005794", "GO:0031090", "GO:0031982" and "GO:0012505". The list of genes localized in the nucleus was obtained from "GO:0005634".

## Identify the regulatory links between TFs and their target genes

The links between TFs and their target genes were first identified based on co-expression analysis. The list of annotated TFs was obtained from RcisTarget package[22] and human TF database[85]. Only the TFs annotated in both databases were selected. The gene expression at the spliced RNA and unspliced RNA levels were calculated as described above. The normalized expression values were log-transformed, centered, and scaled. For each gene, the correlation coefficients between its expression at the unspliced RNA level and the expression of all TFs at the spliced RNA level were calculated. The TFs with r > 0.15 were selected to construct the LASSO regression model with glmnet package. The LASSO regression model was constructed to predict the unspliced RNA expression of the gene of interest based on the expression of selected TFs. As a negative correlation could be caused by the mutual exclusive gene expression patterns, only a positive correlation was considered here. Different criteria were also evaluated, and the majority of TF-gene links can be reproducibly captured with the parameters within the appropriate interval (Supplementary Note 4 and Supplementary Fig. 21). The links with regression coefficients >0.03 were considered. The linked genes of each TF were then compared to Type I kinetic modules. The significant enrichment was determined by FDR < 0.01, enrichment fold >2, and the number of overlapped genes >10. The enriched TF-gene links were then subjected to ChIP-seq verification or motif enrichment analysis.

## Establishing TF regulatory network

The links between different TFs were identified based on the expression covariance as described above. All of the identified links were then verified by published ChIP-seq data. The verified TF−TF links were used to establish the TF regulatory network with Cytoscape[86]. For the TF association network in Fig. 5j, all TF−TF links were used.

## ChIP-seq analysis and motif enrichment analysis

The peak files were downloaded from CistromeDB[21]. The samples with median mapping quality ≥25, unique mapping rate ≥0.6, PCR bottleneck coefficient ≥0.8, a fraction of reads in peaks ≥0.01, peak number (fold > 10) ≥150, and high consistency with DNase-seq data (percentage of top 5k peaks overlapped with DNase-seq ≥0.85) were kept. The rest of the datasets were not used due to potentially low quality. The qualified datasets were then grouped based on their target TFs. For each TF, the peaks (fold > 10) identified in corresponding datasets were merged, and the peak annotation was performed using HOMER[87]. The binding targets were identified if the peak was within 1 or 5 kb from the transcription start site. These two different cutoffs corresponded to the scenarios of binding to the promoter region or binding to the enhancer region, respectively.

For ChIP-seq peak visualization, the raw data or the processed data were downloaded. The processed data were directly used for visualization in Integrative Genomics Viewer (IGV)[88]. Raw data were mapped to human genome assembly (GRCh37) using Bowtie2[89] after removing the adapter sequences with Cutadapt. After deduplication, BAM files were converted to bigwig files using bamCoverage function in deepTools package[90]. Bigwig files were then used for visualization in IGV.

Motif enrichment analysis was carried out within 500 bp upstream of TSS or 5 kb around the TSS using RcisTarget package[22]. The significant enrichment in motif analysis was determined by normalized enrichment score (NES) >3 and the number of enriched genes ≥10.

## Identify TF-gene links using GENIE3

The catalog of human TFs was acquired as described above. Next, gene expression at the spliced RNA and unspliced RNA levels were normalized and log-transformed. An expression matrix consisting of the exon expression data for TFs and intron expression data for the rest of genes was constructed, which was used as the input for GENIE3 algorithm. GENIE3-based co-variation analysis was performed using

SCENIC R package[22]. The TF-gene links with positive correlation (with "corrthr = 0.05" option) and with weight falling within the top 5% are considered in the following analysis. The linked genes of each TF were then compared to Type I kinetic modules, and the significant enrichment was determined as described above.

### Gene knockdown with CRISPR interference (CRISPRi)

HEK293T-CRISPRi cell line was established by using dCas9-KRAB-MeCP2 plasmid (addgene #122205). The cells that were successfully transduced were selected with 10 μg/mL Blasticidin (InvivoGen) for a week. The sgRNA vector was generated based on CROPseq-Guide-Puro (addgene #86708). To enable the measurement of the cell population transduced with sgRNA vector via fluorescence, we replaced the puromycin resistance gene (PuroR) with RFP. Briefly, the CROPseq-Guide-Puro vector was digested by using MluI (NEB) and SmaI (NEB), which was then ligated with the RFP sequence from pLKO5.sgRNA.EFS.tRFP plasmid (addgene #57823). To clone the sgRNA sequence into the sgRNA vector, the sgRNA vector was digested by using Esp3I (ThermoFisher), and the digested vector was next ligated with the annealed sgRNA oligo by using T4 ligase (New England BioLabs). The lentivirus was prepared, and the sgRNA was then transduced into the HEK293T-CRISPRi cell line. After 3 days of transduction, the transduced cells were 1:1 mixed with uninfected HEK293T-CRISPRi cells. The rest of the cells were harvested for RNA extraction, and qRT-PCR was performed to verify the knockdown efficiency by using *ACTB* as the internal control. After 5 days of transduction, the initial percentage (referred to as day 0) of RFP⁺ cells in the mixed cell population was measured by using flow cytometry. The mixed cell population was further cultured for 14 days, and the percentage of RFP⁺ cells was measured by using flow cytometry after 14 days of culturing to determine the effects of the target gene on cell proliferation. The relative proliferation was calculated as RFP⁺% (day 14)/RFP⁺% (day 0). The sgRNA sequences are provided in Supplementary Data 11.

### Data analysis of the gene expression dynamics during oncogene-induced senescence

After mapping and UMI counting, lowly expressed genes were filtered out, and normalization was performed as described above. PCA was carried out with the top 1000 most variable genes. The first five principal components were selected to generate UMAP. Cell cycle analysis was performed on the cells collected at each time point respectively by using reCAT. RNA velocity analysis was performed by using the scVelo package. The raw count matrices were used as the input. To filter the lowly expressed genes, the "min_shared_counts" was set as 50 and the "min_cells_u" as 10. After normalization and log-transformation, the velocities were projected by using the top 2000 most variable genes and the "dynamical modeling" mode with the following parameters: n_pcs = 5, n_neighbors = 20. G0 cells were then ordered based on the latent time to establish the trajectory toward oncogene-induced senescence. The differential gene expression analysis was performed based on the established trajectory by using tradeSeq as described above. The Gene Ontology enrichment analysis was performed on each kinetic module by using "Hypergeometric" model in "goseq" R package.

### Statistics & reproducibility

All statistical analyses were performed using R (4.1.0). All in vitro experiments were repeated at least three times unless otherwise stated. No data were excluded from the analyses. Comparisons between the two groups were performed by using unpaired two-tailed Student's *t*-tests. ANOVA tests were performed for the comparisons between three or more groups. The statistical significance of enrichment analysis was determined by using one-tailed Fisher's exact test. Benjamini–Hochberg procedure was used for multiple testing correction. The details on statistical analyses, tests used, size of *n* and definition of significance are included in the corresponding figure legend and the "Results" section.

### Reporting summary

Further information on research design is available in the Nature Portfolio Reporting Summary linked to this article.

## Data availability

The raw data and the processed datasets generated in this study have been deposited in GEO under the accession number GSE202126. The raw data of Smart-seq3 were obtained from EMBL-EBI ArrayExpress under the accession number E-MTAB-8735. The raw data of CEL-Seq2 were obtained from GEO under the accession number GSE132044. The raw data of the VASA-plate were obtained from GEO under the accession number GSE176588. The raw data of Smart-seq-total were obtained from GEO under the accession number GSE151334. The public ChIP-seq data were obtained from GEO under accession number GSM2132552 for E2F1, GSE59703 for KLF11, GSE55105 for REL and NFKB2, GSE71848 for GTF2B, GSE43036 for STAT1, GSE69309 for ATF4, and from ENCODE under the accession number ENCFF826PYA for E2F2, ENCFF487KQK for MYBL2, ENCFF437PEU for KLF10, ENCFF133YDB for STAT2, ENCFF937HON for ELF1, ENCFF783IOJ for ATF3, ENCFF182SSK for CEBPG. Source data are provided with this paper.

## Code availability

The analysis pipeline customized for snapTotal-seq sequencing data is available at https://github.com/zonglab/snapTotal-seq.git[91].

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

## Acknowledgements

We are grateful to the McNair family for their support. We would like to thank other Zong Lab members for their kind support. C.Z. is supported by a McNair Scholarship, NIH Director's New Innovator Award (Grant No.1DP2EB020399), the Behavioral Plasticity Research Institute (Grant No. DBI-2021795), and NIH SMaHT program (Grant No. 1UG3NS132132). FACS experiments were performed in the Cytometry and Cell Sorting Core at Baylor College of Medicine with funding from the CPRIT Core Facility Support Award (No. CPRIT-RP180672), the NIH (Grant Nos. P30 CA125123, S10 RR024574, and S10OD025251) and the assistance of J.M. Sederstrom.

## Author contributions

C.Z. and Y.N. designed the project. Y.N. and J.L. performed the experiments. Y.N. and C.Z. performed data analysis. C.Z. and Y.N. wrote the manuscript.

## Competing interests

C.Z. is a co-founder and equity holder of Pioneer Genomics Inc. Other authors declare no competing interests.
