## [Peer Review File · Nature Communications]

Single-cell Total-RNA Profiling Unveils Regulatory Hubs of Transcription FactorsReviewers' Comments:

Reviewer #1:

None

Reviewer #2:

Remarks to the Author:

In Niu et al, the authors present a new single-cell total RNA sequencing method in which MALBAC-based primers are used to initiate RT at low temperature in single cells. Taking advantage of the recovery of mature (spliced) and nascent (unspliced) transcripts with single cell resolution, they utilize RNA velocity to identify modules of gene expression that undergo different regulatory mechanisms during cell cycle and oncogene-induced senescence (OIS). The analysis presented is creative and sustained in reasonable hypothesis. However, the description and benchmarking of the technology is at times a bit superficial.

In what follows, I split my comments according to whether they target the technological, cell cycle, or OIS aspect of the paper. I hope the authors find it useful.

Technology:

- As the field of transcriptomics realizes of the power of total RNA interrogation, some definitions need to be standardized. One of these is precisely "total RNA". The authors define total RNA as the sum of unspliced and spliced protein coding RNA. However, for other researchers, total RNA is the combination of all the RNA biotypes (protein coding, long non-coding, short non-coding). I encourage the authors to make this distinction in the introduction of their work and add an analysis to show which biotypes their technology is able to detect. Do they see miRNA, snRNA, lncRNA, etc? With that in mind, maybe the authors could rephrase a bit line 109, regarding the primary advantage of total-RNA based methods.
- The results from the species-mixing experiment should be reported differently: if the authors performed the library prep combining cells from different species simultaneously, a scatter plot showing the number of reads assigned to human versus number of reads assigned to mouse must be provided. This allows the reader to assess the risks of doublets and "transcript hopping" from cell to cell. If the authors did not perform the experiment as such, then they should remove all the species-mixing results from the paper.
- In lines 103-104, the authors report the number of reads as 10372 \pm 518 and 10431 \pm 875. First, they should define what the error is. Second, this is not how errors are reported. The authors should express these results using the appropriate number of significant digits (10400 \pm 500 and 10400 \pm 900, respectively). Check the book "An introduction to Error Analysis", from John R. Taylor, for example.
- In line 85, the authors could also refer to Sup. Fig. 2 as well.
- To investigate batch effects, the authors perform two batches of snapTotal-seq using HEK293T cells. Are these biological, or technical replicates?
- According to the snapTotal-seq protocol, the cell barcode is introduced after 20 cycles of PCR amplification. Does this have any advantage? I imagine that the authors could also introduce the cell barcode during the PCR amplification itself. Can the authors comment on this?
- In Sup. Fig. 1 and Sup. Fig. 2, primer descriptions need to be improved. For example, in Sup. Fig. 1, the UMI should be made more visible and the figure caption should describe what the red the blue and the green sequences are. The same applies for Sup. Fig. 2.
- Could the authors clarify if snapTotal-seq maintains transcript strandness? If it does not, which I think is the case, a column regarding "Strandness" should be added to the table of Sup. Fig. 1.
- Which modification do we need to perform in order to make the protocol compatible with the Truseq or the Nextera primers, instead of using snapTotal-seq-specific primers?
- Does snapTotal-seq have the potential to become a droplet-based method?
- In Sup. Fig. 4a, why do smartseq3 and celseq2 perform worse at the exon detection level? Did the authors "correct" for number of reads in order to perform this benchmarking analysis?

- In the methods, the authors should also provide the sequence of the DSC primer with cell barcode.

Cell cycle and RNA velocity

- Can the authors detect how histones and other transcript biotypes behave during cell cycle?
- Information about module assigned and logFC should be added in Sup. Table 1.
- What is the unit of "offset" in Fig. 2b?
- To which modules do the genes validated by qPCR in Fig. 2f-g belong to? Maybe they could be annotated in the heatmaps of Fig. 2b?
- In Fig. 2h, the authors perform a differential fraction of genes analysis between known and unknown CCGs between types I, II and III. However, I do not understand why this analysis on statistical significance is performed here. This also applies to Fig. 4l, what does the FDR stand for there?
- Regarding the analysis in Fig. 2h, are there genes from the list that are missing in the snapTotal-seq?
- 23 TF are identified with potential downstream genes being significantly enriched in Type I kinetic modules. Is this identification based on LASSO regression, or based on target sequences in gene promoters?
- In Fig. 3b, the colorscale indicating the FDC seems to be irrelevant, since all the points have the same color. Is this intentional?
- In Fig 3c, it would be nice to have the 23 TF annotated in the heatmap.
- Also, in Fig. 3e, BPTF, RREB1 and TBP1 are missing. Can the authors speculate on their role during cell cycle?
- In Figure 3g-k there are some points that have high LASSO coefficients but are not annotated. Despite the fact that these points do not belong the TFs from the discussion, it might be interesting for the reader to know what are they. Maybe the authors. Alternatively, the authors could provide a supplementary table reporting the LASSO coefficients for the different study cases.
- I am having problems connecting the analysis from Fig. 3g-k to the reconstructed network in Fig. 3l: from the text, I reason that the take-home message of Fig. 2g (for example) is that changes in mature E2F2 and KLF11 are having an impact on the transcription dynamics of E2F1. Therefore, in Fig. 3l I would expect an arrow foing from KLF11 to E2F1 (which is there) and another from E2F2 to E2F1 (which is there, but in the reverse orientation). Can the authors clarify this?
- In Table S1 it would be helpful to add a column indicating to which module each gene belongs to. The same would apply to table S3.
- A better explanation of the knock-down experiments reported in Supplementary Fig. 8a should be added in the caption of the figure or in the text. As it is, it is not straightforward to understand the RFP+ readout until one reaches the Methods section.
- In Fig. 4l, is that mature or nascent YTHDF2?

Oncogene-induced senescence

- The authors sometimes report results using "cell trajectory" and others using "latent time". Is that the same? How does the latent time relate to the physical time (day 1, 2,3 and 5) of the experiment? Does the latent time recapitulate well the physical time? What is the top bar in Supp. Fig. 10a?
- In Table S3, it could be useful to add a column reporting whether the gene is a TF and to which Hub it is assigned.

Conclusions

- In lines 48-50, the authors claim that total-RNA seq methods without the need for chemical labeling are more suited for kinetic and regulatory analysis. Do they have the evidence to claim this? Chemical labeling results should be included in the paper. I would recommend the authors to remove this sentence or, alternatively, to add a section on metabolic labeling during cell cycle and benchmark.

Reviewer #3:

Remarks to the Author:

The manuscript "Single-cell total RNA profiling reveals regulatory hubs of transcription factors" by Niu et al present a novel sequencing approach, snapTotal-seq, illustrated several examples, highlighting the improvements in detecting regulatory hubs.

One first remark is that the manuscript would benefit from a more balanced distributions of sections; a lot of effort went into the results section, yet the discussion is somewhat minimal. Perhaps the authors could expand on this latter part, and include some of the cross comparisons with other approaches there.

The second generic comment is that an increase in the level of details (both for the description of methods and the information presented on various panels) would most certainly assist the readers. e.g. PCA visualisations are rarely informative for large expression matrices (large number of genes and large number of samples/ cells). the approach is suitable for bulk/ mini-bulk experiments, but at single cell level needs to be supported with a % variance explained; if a low proportion of variance is explained, then non-linear dimensionality reduction approaches (such as UMAP, used later on) should be employed. Another example of insufficient information is the sentence from the abstract "with lasso regression between transcription factors we identified the critical regulatory hubs" - regression on its own, even if a lasso (or even elastic net) regularisation, is unlikely to capture the complex covariation between genes (also, correlation analyses are not appropriate in this context!); please consult the literature proposing machine learning or non-linear approaches for characterising co-variation.

A third generic comment is related to the mixture of input data (e.g. RNA/ChIP) without sufficient details that could underline the biological link and compatibility of those datasets.

"Interestingly, we noticed Smart-seq-total has a low gene detection with either exon or intron reads, which is potentially due to the dominance of detection of miscRNA in their data, suggesting that this method is more suited for detecting short RNA species."

The miscRNA class is (at least partially) included in the definitions of genes.

To ensure the clarity of text, please indicate whether by gene you understand protein-coding gene, or transcribed RNA. Moreover, the short RNAs, if you are referring to small RNAs, are distinct from the miscRNA class.

134-138 although I fully recognise the importance of visualisation, conclusions should not be drawn solely on visualisations. Moreover, the circular distribution of points, might suggest the horseshoe effect.

142-143 it is incorrect to rely on the correlation-coefficient to compare the two approaches.

I also think that a comparison across methods from the perspective of velocity directionality and amplitude would increase the impact of the manuscript.

150-153 the selection of DE genes should not be based on FDR alone (using the relaxed 0.1 threshold should be justified, since it exposes the analysis to potential false positives). Especially with a relaxed threshold, a fold change criteria should be introduced. It would be interesting to see the numbers of genes of interest when stricter criteria (including one on abundance) are introduced.

182-184 it is interesting to see such a large proportion of genes missed by existing analyses.

I would like to see some more supporting information that these are not false positives.

Also, are there any canonical cell cycle genes, that are present in the data, but not detected by your analysis?

201-203 as mentioned previously, include some cross comparison with other non parametric approaches.

220- 222 I fully support the inclusion of other modalities. However, it is well known that slight experimental changes can induce significant modifications in analyses. You could use e.g. SCENIC on the RNA expression, infer some regulons, and then compare and contrast the results using the publicly available ChIP data.

264-266 clarify what you mean by cross correlation analysis

324 - 336 it is unclear why you suddenly change dataset. Even though the manuscript is presented as technical, some of the partial biological conclusions confuse the message. Moreover, it is unclear why some of analyses steps described for the first dataset are recapitulated on the second dataset (345-393). If possible, I recommend focusing on one dataset, exemplifying the characteristics of the assay from all required angles, in a focused manner.

Minor comments

The main figures were not clearly labelled making it difficult to follow the information. I recommend presenting the code used to generate these results in a github repository. Moreover, if apps could be created to illustrate the dataset(s), it would be very useful for the community.

Reviewer #4:

Remarks to the Author:

The manuscript entitled "Single-cell Total-RNA Profiling Unveils Regulatory Hubs of Transcription Factors" introduces snapTotal-seq, a promising addition to the growing pool of single-cell total RNA sequencing methods. The method was benchmarked against multiple single-cell RNA-seq assays for kinetic analysis of the cell cycle. The manuscript applies LASSO regression to discern transcription factor regulatory hubs that orchestrate cell cycle dynamics. Importantly, snapTotal-seq's ability to quantify total RNA at a single-cell level is a significant contribution to single-cell transcriptomic methodologies. I recommend publishing this manuscript with a few improvements to substantiate its findings and enhance reproducibility.

1. The manuscript asserts that snapTotal-seq concurrently quantifies nascent and mature RNA. However, it is crucial to delineate that "nascent" RNA refers to transcripts undergoing synthesis. The RNA species captured by this method and analyzed in this manuscript are unspliced and spliced RNA. Given that intron retention and detention are known regulatory phenomena, it is essential to differentiate and accurately label unspliced or "intronic" RNA from "nascent" RNA. Throughout the manuscript, the term "nascent" should be replaced with either "unspliced" or "intronic".
2. For a robust assessment of snapTotal-seq, I recommend a more quantitative approach in the benchmarking analyses. Clarity on the number of cells analyzed, selection criteria for cells, thresholds and parameters for cell inclusion, number of cells in Figure 1c, the proportion of intronic read and improvement in unspliced RNA capture efficiency over polyA-capture methods, and the distribution of read coverage across gene loci would provide a clear picture of the method's capabilities.
3. A quantitative comparison of snapTotal-seq's performance against other single-cell RNA-seq methodologies in terms of dynamic range and bias towards RNA species would further improve clarity. Similarly, evaluation of the non-coding RNA, such as the proportion of reads from regulatory regions and whether snapTotal-seq has a higher capture efficacy for long RNA over short, would help in gauging whether the method truly and unbiasedly captures all RNA.

4. The choice of nucleotides for annealing and reverse transcription is surprising. Would a widely used random hexamer (NNNNNN), as opposed to T12, T3, and G3 sequences used in this method, mitigate potential read biases, particularly towards the 3'-end of genes?

5. The number of cells in other assays used in benchmarking was relatively low (100 – 200). Is this the total number of cells in these assays, or a small number of cells were randomly sampled to match the number of cells in snapTotal-seq? If it was sampled, what was the threshold/criteria?

6. Concerning cell cycle analysis, the process for deriving five gene modules in Figures 2b, 4a, and 5f remains unclear. It would be beneficial to elucidate the clustering algorithm employed and provide module gene counts.

7. The authors validated known CCGs and novel CCGs for Type I genes in Fig. 2f and 2g. Similar qRT-PCR-based validation for a handful of Type II and Type III genes would strengthen the robustness of CCGs classification. Such validation becomes further critical in the context of the claim that 80-90% of Type II and Type III genes do not have known cell cycle functions (Fig. 2h).

8. It is essential to clarify whether the novel CCGs detected are unique to snapTotal-seq or simply overlooked or undetectable by other total-RNA-capturing single-cell methods. The authors could bolster the superiority of total-RNA-based methods by showing evidence of similar cell-cycle dependence of these novel CCGs in other single-cell total-RNA methods.

9. The authors presented the post-transcriptional regulatory mechanisms governing Type II genes. However, I am not convinced that the evidence provided in Figure 4h substantiates the claim of a correlation between the expression patterns of four RNA-binding proteins and their respective target gene groups throughout the cell cycle. Should the RBP act to stabilize its bound RNA, one would anticipate a mirrored expression pattern between the target unspliced RNA and the RBP. Conversely, if the RBP serves to destabilize the RNA, their expression patterns should be inversely related. The Figure 4h does not seem to provide this evidence.

10. Lastly, a thorough exposition of the LASSO regression model, including parameter choices and model validation, would enhance interpretability.

On a minor note, the current layout of figures, with their numerous panels, tends to obscure the central findings. Transferring some of these panels to the supplementary material would help maintain focus on the most significant results.

We would like to thank all the reviewers for their comments and suggestions. Point-to-point responses are provided below.

REVIEWER COMMENTS

Reviewer #2 (Remarks to the Author):

In Niu et al, the authors present a new single-cell total RNA sequencing method in which MALBAC-based primers are used to initiate RT at low temperature in single cells. Taking advantage of the recovery of mature (spliced) and nascent (unspliced) transcripts with single cell resolution, they utilize RNA velocity to identify modules of gene expression that undergo different regulatory mechanisms during cell cycle and oncogene-induced senescence (OIS). The analysis presented is creative and sustained in reasonable hypothesis. However, the description and benchmarking of the technology is at times a bit superficial.

In what follows, I split my comments according to whether they target the technological, cell cycle, or OIS aspect of the paper. I hope the authors find it useful.

Technology:

- As the field of transcriptomics realizes of the power of total RNA interrogation, some definitions need to be standardized. One of these is precisely “total RNA”. The authors define total RNA as the sum of unspliced and spliced protein coding RNA. However, for other researchers, total RNA is the combination of all the RNA biotypes (protein coding, long non-coding, short non-coding). I encourage the authors to make this distinction in the introduction of their work and add an analysis to show which biotypes their technology is able to detect. Do they see miRNA, snRNA, lncRNA, etc? With that in mind, maybe the authors could rephrase a bit line 109, regarding the primary advantage of total-RNA based methods.

We thank the reviewer for his or her critiques here. Indeed, we are able to detect all RNA biotypes in addition to protein coding genes, as shown below. To improve the precision of our definition, we have rephased the introduction part and the description at line 109 as suggested by the reviewer. The corresponding figure was added to Supp. Fig. 3c.

- The results from the species-mixing experiment should be reported differently: if the authors performed the library prep combining cells from different species simultaneously, a scatter plot showing the number of reads assigned to human versus number of reads assigned to mouse must be provided. This allows the reader to assess the risks of doublets and “transcript hopping” from cell to cell. If the authors did not perform the experiment as such, then they should remove all the species-mixing results from the paper.

We apologize for the potential confusion. Yes, we have performed the standard species-mixing experiment by combining cells from different species simultaneously first followed by the snapTotal chemistry. The scatter plot is provided in **Supp. Fig. 3a** as suggested by the reviewer.

- In lines 103-104, the authors report the number of reads as 10372+/-518 and 10431+/-875. First, they should define what the error is. Second, this is not how errors are reported. The authors should express these results using the appropriate number of significant digits (10400+/-500 and 10400+/-900, respectively). Check the book "An introduction to Error Analysis", from John R. Taylor, for example.

We apologize for the confusion here. At the bottom of page 5, we have modified to show the correct form as suggested.

- In line 85, the authors could also refer to Sup. Fig. 2 as well.

The reference to Sup. Fig2 has been added.

- To investigate batch effects, the authors perform two batches of snapTotal-seq using HEK293T cells. Are these biological, or technical replicates?

We have performed technical replicates. We have modified our description to clarify this.

- According to the snapTotal-seq protocol, the cell barcode is introduced after 20 cycles of PCR amplification. Does this have any advantage? I imagine that the authors could also introduce the cell barcode during the PCR amplification itself. Can the authors comment on this?

We thank the reviewer for this detailed design question. Yes, it does have major advantage. We would like to point out that the adapter sequences on both ends of the amplicon are identical. If we introduce the cell barcode during PCR amplification step, it would result in the cell barcode being added to both ends of the amplicon products, as a result, it would create a substantial stem loop structure that essentially hinders the efficient amplification. In contrast, when we introduce cell barcode via the extra double strand conversion step after the PCR amplification, the cell barcode are only be added to one end of the amplicon. We add this paragraph description in the METHODS (the middle part of Page 32) to facilitate the understanding of this chemistry design.

- In Sup. Fig. 1 and Sup. Fig. 2, primer descriptions need to be improved. For example, in Sup. Fig. 1, the UMI should we made more visible and the figure caption should describe what the red the blue and the green sequences are. The same applies for Sup. Fig. 2.

We appreciated the reviewer's comments here. We have improved the resolution of those figures and the description of the figures in the captions.

- Could the authors clarify if snapTotal-seq maintains transcript strandness? If it does not, which I think is the case, a column regarding “Strandness” should be added to the table of Sup. Fig. 1.

Yes, the reviewer is correct in his or her understanding of our chemistry. snapTotal-seq does not maintain transcript strandness. We also added a new column into the table in Supp. Fig.1b to clarify this feature.

- Which modification do we need to perform in order to make the protocol compatible with the Truseq or the Nextera primers, instead of using snapTotal-seq-specific primers?

We apologize that our description for converting the PCR products to TruSeq based sequencing library is less clear. We have improved our description of METHODS at bottom of Page 33. In brief, the snapTotal-seq libraries can be easily converted to Illumina standard libraries using one step of PCR by using a primer with Illumina TruSeq adapter sequence in the final PCR step.

- Does snapTotal-seq have the potential to become a droplet-based method?

We greatly appreciate the suggestions provided by the reviewer here. We are currently working on adapting snapTotal chemistry to the droplet-based platform. It is worth noting that we successfully adapted MATQ-seq to a droplet-based platform (MATQ-drop) for profiling the total-RNA based transcriptome in fixed nuclei and subcellular structures¹. With this experience, we believe the adaption of snapTotal chemistry should be relatively straightforward.

- In Sup. Fig. 4a, why do smartseq3 and celseq2 perform worse at the exon detection level? Did the authors “correct” for number of reads in order to perform this benchmarking analysis?

Yes, to achieve an equal footing comparison, we have downsampled the cells sequenced by different methods to the same depth (1M reads per cell) in the benchmark analysis. We have added this description into caption to make it clear in the Sup. Fig. 5. The lower gene detection by Smart-seq3 and CEL-Seq2 at the exon level is likely attributed to the less efficient transcript capture by using oligo-dT primer only.

- In the methods, the authors should also provide the sequence of the DSC primer with cell barcode.

We apologize for missing this detail in the previous version. The sequence of the DSC primer has been added to the METHODS.

Cell cycle and RNA velocity

- Can the authors detect how histones and other transcript biotypes behave during cell cycle?

Yes, we can detect the gene expression changes in histones and other transcript biotypes (in total 34 noncoding genes at the exon level) along cell cycle. The cell-cycle dependent

expression of histone genes and the examples of other RNA biotypes are presented in Supp. Fig. 4d-e in the violin plot format.

- Information about module assigned and logFC should be added in Sup. Table 1.

We apologize for missing this detail in the previous version. The related information has been included in Supplementary tables (as Table S2 and S3) as suggested by the reviewer here.

- What is the unit of “offset” in Fig. 2b?

We thank the reviewer for pointing this out. The unit used in Fig. 2b represents a time unit, where one single unit corresponds to $\frac{\text{cell division time}}{\text{number of cells}}$. To facilitate the understanding of this pseudotime unit, we changed the unit to the percentage of cell cycle in the revised manuscript.

- To which modules do the genes validated by qPCR in Fig. 2f-g belong to? Maybe they could be annotated in the heatmaps of Fig. 2b?

These genes indeed belong to different modules since we randomly chose genes for qPCR validation. As suggested by the reviewer here, we have added the verified genes to the heatmap in Fig. 2b to facilitate the easy reading.

- In Fig. 2h, the authors perform a differential fraction of genes analysis between known and unknown CCGs between types I, II and III. However, I do not understand why this analysis on statistical significance is performed here. This also applies to Fig. 4l, what does the FDR stand for there?

We apologize for missing the details of statistical tests. in the previous version. The analysis was done with Fisher’s Exact test. We added this information in the caption of Fig. 2h. In Fig. 2h, we would like to examine the distinct enrichment patterns of known CCGs across three types of CCGs identified by our DEG analysis. The observed differential enrichment indicated that known CCGs were mainly regulated at the transcriptional level, whereas the unknown CCGs could be regulated at both transcriptional (the ones belonging to Type I) and post-transcriptional (the ones belonging to Type II) levels. We have added the description of the statistics test in the caption.

In Fig. 4l, the FDR stands for the statistical significance of the differential expression of YTHDF2 along cell cycle. We have added the description of the statistics test in the caption.

- Regarding the analysis in Fig. 2h, are there genes from the list that are missing in the snapTotal-seq?

Yes, there are genes from the known cell cycle gene list that are not captured by our data. Firstly, the genes regulated at the protein level (e.g., protein modification, interacting with inhibitory protein) throughout the cell cycle were not detected by our analysis. For instance, cyclin D (CCND1, CCND2, CCND3), which drives G1/S transition, undergoes tight regulation by ubiquitin-proteasomal pathway along cell cycle^{2,3}, and no significant changes at the RNA level were detected in our data. Similarly, CDK4/6 is activated through the assembly with cyclin D and is inhibited by the binding of inhibitory proteins from INK4 family or CIP/KIP family⁴⁻⁶. Consistently, no significant changes at the RNA level were detected in our data for these two genes.

Secondly, previous studies have shown that different regulators were involved in the regulation of cell cycle in different cell types^{5,7,8}. Since the list of known cell cycle gene list is curated from previous studies based on a variety of biological contexts, it could include cell-type specific cell cycle regulators which may not participate in the cell cycle regulation in HEK293 cell line we characterized here.

We have added this part of description into the discussion section (the bottom paragraph in Page 23) to show the full spectrum of our analysis.

- 23 TF are identified with potential downstream genes being significantly enriched in Type I kinetic modules. Is this identification based on LASSO regression, or based on target sequences in gene promoters?

Thanks for question. The identification of potential downstream genes is based on LASSO regression.

- In Fig. 3b, the colorscale indicating the FDC seems to be irrelevant, since all the points have the same color. Is this intentional?

We apologize for the confusion here. As the FDR values of the identified enrichment are all < 0.01, it was hard to tell the differences in color using the previous color scale. In the revised manuscript, we have modified the color scale in Fig. 3b.

- In Fig 3c, it would be nice to have the 23 TF annotated in the heatmap.

We thank the reviewer for this suggestion. These 23 TFs are annotated in the revised manuscript as suggested by the reviewer here.

- Also, in Fig. 3e, BPTF, RREB1 and TBP1 are missing. Can the authors speculate on their role during cell cycle?

We thank the reviewer for carefully assessing our result and the suggestion.

The three genes are missing because the regulatory relationships between other TFs and them were not identified through the co-variation analysis. This result suggests that these three genes might regulate gene expression along cell cycle via different mechanisms, independent of the TF network studied here. For example, BPTF, as a core subunit of NURF chromatin complex, has been reported to function as a co-factor of MYC to regulate gene expression⁹. Therefore, the regulatory effects of BPTF on its downstream genes could depend on the activities of other TFs, e.g., MYC. RREB1 is a transcription factor that acts downstream of RAS signaling pathway, which binds to RAS responsive elements upon the activation of RAS signaling¹⁰. As a result, the activity of RREB1 could depend on RAS signaling pathway during cell cycle. TBP1 encodes the TATA-binding protein, which is an important subunit of the basal transcription factor complex. Previous studies have also shown that TBP1 closely associates with condensed chromosomes during mitosis, and facilitates the transcriptional reactivation after mitosis¹¹⁻¹³. Therefore, TBP1 could play an important role to coordinate the general transcriptional activity and the chromosome condensation and segregation during mitosis.

We have added this description to our revised manuscript to provide this discussion (the top paragraph on Page 28 and Supplementary text 1).

- In Figure 3g-k there are some points that have high LASSO coefficients but are not annotated. Despite the fact that these points do not belong the TFs from the discussion, it might be interesting for the reader to know what are they. Maybe the authors. Alternatively, the authors could provide a supplementary table reporting the LASSO coefficients for the different study cases.

We thank the reviewer for the thoughtful suggestions here. As suggested, we added a supplementary table (Supp. Table 5) to report the TFs with high LASSO coefficients.

- I am having problems connecting the analysis from Fig. 3g-k to the reconstructed network in Fig. 3l: from the text, I reason that the take-home message of Fig. 2g (for example) is that changes in mature E2F2 and KLF11 are having an impact on the transcription dynamics of E2F1. Therefore, in Fig. 3l I would expect an arrow foing from KLF11 to E2F1 (which is there) and another from E2F2 to E2F1 (which is there, but in the reverse orientation). Can the authors clarify this?

First, we would like to confirm that the reviewer's interpretation is correct. And we apologize for the confusion. In Fig. 3l, there are indeed two arrows pointing to E2F1. One is from KLF11 and the other one is from E2F2. However, as several lines were mingled together in this plot, it is hard to tell them apart. In the revised manuscript, we have modified the figure to improve the clarity.

- In Table S1 it would be helpful to add a column indicating to which module each gene belongs to. The same would apply to table S3.

We thank the reviewer for the suggestions. The information regarding gene modules has been added as Supplementary Table 3 for cell cycle study and Supplementary Table 7 for OIS study in the revised manuscript.

- A better explanation of the knock-down experiments reported in Supplementary Fig. 8a should be added in the caption of the figure or in the text. As it is, it is not straightforward to understand the RFP+ readout until one reaches the Methods section.

We thank the reviewer for the suggestions. We have modified the caption of Supplementary Fig. 15a (the Supp. Fig 8a in the previous version). to make it easy to read.

- In Fig. 4l, is that mature or nascent YTHDF2?

It is the mature RNA level of YTHDF2. We improved the description in the caption to make it clear.

Oncogene-induced senescence

- The authors sometimes report results using “cell trajectory” and others using “latent time”. Is that the same? How does the latent time relate to the physical time (day 1, 2,3 and 5) of the experiment? Does the latent time recapitulate well the physical time? What is the top bar in Supp. Fig. 10a?

Yes, “Cell trajectory” and “latent time” are closely related. We use “Cell trajectory” to refer the positioning on the cells in phase diagram and we use latent time to refer the time at which the cell is at certain position of trajectory. We apologize for the mixed usage of these terms.

The physical time was well recapitulated by the latent time in our data. In the revised manuscript, we added the correlation between the latent time and the physical time in Supp. Fig 16d.

The top bar in Supp Fig. 10a in the original manuscript represents the physical time (day 1, 2, 3 and 5) of each cell. To be consistent with other related plots (Fig. 5f), we have changed the top bar in this plot to “latent time” in the revised manuscript.

- In Table S3, it could be useful to add a column reporting whether the gene is a TF and to which Hub it is assigned.

We would like to thank the reviewer for the suggestions here. To make it easier to read, we added a new table (Table S8) to report the identified TF hubs in the revised manuscript.

Conclusions

- In lines 48-50, the authors claim that total-RNA seq methods without the need for chemical labeling are more suited for kinetic and regulatory analysis. Do they have the evidence to claim this? Chemical labeling results should be included in the paper. I would recommend the authors to remove this sentence or, alternatively, to add a section on metabolic labeling during cell cycle and benchmark.

We thank the reviewer for this suggestion to strengthen the rigidity of our claim. As suggested, we have removed the related statements in the revised manuscript.

Reviewer #3 (Remarks to the Author):

The manuscript "Single-cell total RNA profiling reveals regulatory hubs of transcription factors" by Niu et al present a novel sequencing approach, snapTotal-seq, illustrated several examples, highlighting the improvements in detecting regulatory hubs.

One first remark is that the manuscript would benefit from a more balanced distributions of sections; a lot of effort went into the results section, yet the discussion is somewhat minimal. Perhaps the authors could expand on this latter part, and include some of the cross comparisons with other approaches there.

We appreciate the reviewer's advice here, and we have tried to improve the balance of materials a bit in the revised manuscript. But it is difficult to achieve that. We did not move the cross comparison to the end as suggested by this reviewer, since we feel the benchmark comparison is important for establishing the credibility of the method before we can dive into more detailed biological studies.

The second generic comment is that an increase in the level of details (both for the description of methods and the information presented on various panels) would most certainly assist the readers. e.g. PCA visualisations are rarely informative for large expression matrices (large number of genes and large number of samples/ cells). the approach is suitable for bulk/ mini-bulk experiments, but at single cell level needs to be supported with a % variance explained; if a low proportion of variance is explained, then non-linear dimensionality reduction approaches (such as UMAP, used later on) should be employed.

To address this issue, % variance explained is provided in the corresponding panels in the revised manuscript. We also plotted the 'elbow plot' to determine the principle components that captured the majority of true biological signal, as suggested in Seurat package. In our data, the top 6 PCs captured the majority of true signals based on this analysis. Among them, PC1 and PC2 explained a significant proportion of the total variance (33.55% and 21.42% respectively), which agrees with the observations that cells from different cell cycle stages can be clearly separated by PC1 and PC2 in Figure 1c. The similar analysis was also applied to other datasets in Figure 1.

We thank the reviewer for this suggestion, including more information (i.e., PCs) could provide more details on the biological features. We have performed non-linear dimensionality reduction on the first 6 PCs and plotted the resulting UMAP for visualization. Interestingly, through unsupervised clustering, we were able to identify five cell cycle sub-stages in our data, as evidenced by the expression of a set of well-known cell cycle marker genes. These new results were included in Supp. Fig. 4 in the revised manuscript.

Another example of insufficient information is the sentence from the abstract "with lasso regression between transcription factors we identified the critical regulatory hubs" - regression

on its own, even if a lasso (or even elastic net) regularisation, is unlikely to capture the complex covariation between genes (also, correlation analyses are not appropriate in this context!); please consult the literature proposing machine learning or non-linear approaches for characterising co-variation.

We thank the reviewer for pointing out the limitation of LASSO analysis. To explore other types of analysis as suggested by the reviewer, in the revised manuscript, we employed GENIE3, a non-parametric (model-free) method, for the covariation analysis, and contrasted the results with those obtained from LASSO regression.

To briefly summarize the result, in cell cycle data, 12 out of 23 TF regulons and 67.2% of related TF-gene links identified by LASSO regression were also detected by GENIE3. In senescence data, 78.6% of TF regulons and 81.8% of related TF-gene links identified by LASSO regression were also detected by GENIE3. In addition, the shared TF regulons and TF-gene links have significant higher weights/coefficients than the ones identified by only LASSO regression or GENIE3. These results confirm most of the high confident TF-gene links that have been identified by LASSO regression in our original analysis.

GENIE3 indeed discovered more TF-gene links in both datasets, indicating that the non-parametric approaches are likely more sensitive to detect the complex covariation between genes. We have added these detailed results in Supp Fig. 11 & 17 in the revised manuscript. The discussion regarding the selection of covariation analysis approaches was also added in the discussion section in the revised manuscript, as another way to balance our materials as suggested in the first critique.

A third generic comment is related to the mixture of input data (e.g. RNA/ChIP) without sufficient details that could underline the biological link and compatibility of those datasets.

We thank the reviewer for this generic critique. First, our input data are obtained from the published CistromeDB, in which the source of experimental data has been provided by the authors of this database. In terms of compatibility between RNA-seq and ChIP, we believe that this approach follows the common practice since we generally believe that the cell lines are relatively stable at the mass-action or bulk level. The analysis across different cell lines allows us to examine the generality of our results.

"Interestingly, we noticed Smart-seq-total has a low gene detection with either exon or intron reads, which is potentially due to the dominance of detection of miscRNA in their data, suggesting that this method is more suited for detecting short RNA species."

The miscRNA class is (at least partially) included in the definitions of genes.

To ensure the clarity of text, please indicate whether by gene you understand protein-coding gene, or transcribed RNA. Moreover, the short RNAs, if you are referring to small RNAs, are distinct from the miscRNA class.

We apologize for unclear description. In the analysis on gene detection sensitivity of each method (Supp. Fig. 5a-b), we included all types of gene, including miscRNA.

In the revision, we further analyzed Smart-seq-total data in detail as suggested. As shown in the figure below, we found that a substantial proportion of reads were mapped to a few genes in Smart-seq-total data (RN7SK and RMRP in exon reads, RPS29 and KIAA0907 in intron reads), which was not observed in other methods. Based on these observations, we concluded that Smart-seq-total might have distinct pattern of bias compared to other methods. Therefore, we excluded it from the following analyses.

In the revised manuscript, we modified the statement to the following “Interestingly, we noticed Smart-seq-total has a low gene detection with either exon or intron reads, which is potentially due to the dominant detection of a few RNA species, specifically, RN7SK, RMRP, RPS29 and KIAA0907.” to provide specific details and avoid the potential confusion.

Caption: (a) The percentage of exon reads mapped to different genes in Smart-seq-total data. (b) The percentage of intron reads mapped to different genes in Smart-seq-total data. (c) The percentage of reads mapped to RN7SK, RMRP, RPS29 and KIAA0907 in the datasets of different methods.

134-138 although I fully recognise the importance of visualisation, conclusions should not be drawn solely on visualisations. Moreover, the circular distribution of points, might suggest the horseshoe effect.

We thank the reviewer for this comment. We apologize that our description may lead to that impression. Here, we would like to clarify that our conclusions were not solely drawn from visualizations. First, as shown in Figure 1o, we also employed “velocity confidence score” to evaluate the consistency of the inferred RNA velocity between neighboring cells. A higher confidence score indicated higher reliability and lower technical noise of the method.

Secondly, to evaluate the accuracy of RNA velocity analysis (i.e., directionality), we also compared the inferred RNA velocity to the cell cycle trajectory generated by reCAT, which is based on the expression of well-known cell cycle markers (Figure 1p). A higher correlation between RNA velocity and the reCAT result indicated the higher accuracy in the inference of velocity directionality.

Regarding the ‘horseshoe’ effect, it is interesting to see the distribution actually looks like a horseshoe. But we would like to emphasize the circular distribution was derived from PCA analysis, which is linear dimension reduction. Thus, the cells on the far-left and far-right are the ones at the left part of G2/M (i.e. early part of G2/M) and the right part of G2/M (i.e. late part of G2/M) respectively in Figure 1c, and these two groups of cells indeed shared similar gene expression profiles at certain extent (e.g., expression of G2 phase genes), instead of being at the opposing ends of the linear continuum represented by PCA components. On the other hand, the gap in the circular distribution in fact indicate a rather quick transition occurred in G2/M phase and with the current sampling size, we did not sample this transition state well.

142-143 it is incorrect to rely on the correlation-coefficient to compare the two approaches. I also think that a comparison across methods from the perspective of velocity directionality and amplitude would increase the impact of the manuscript.

We thank the reviewer for this comment, we believe that our previous description may cause the confusion. We would like to emphasize that the correlation-coefficient is used to examine whether an accurate cell trajectory can be inferred by RNA velocity analysis or reCAT for each method, instead of being used to compare the two algorithms. We have removed “reCAT vs scVelo” in the figure to avoid this comparison.

Regarding to the velocity directionality and amplitude analysis suggested by the reviewer, we would like to point out that we have used the velocity confidence score to compare the performance of different scRNA-seq chemistry. And velocity confidence score is a statistical evaluation of consistent directionality and the amplitude of changes along the direction.

150-153 the selection of DE genes should not be based on FDR alone (using the relaxed 0.1 threshold should be justified, since it exposes the analysis to potential false positives). Especially with a relaxed threshold, a fold change criteria should be introduced. It would be

interesting to see the numbers of genes of interest when stricter criteria (including one on abundance) are introduced.

We agree with the reviewer that using a relaxed threshold might increase the likelihood of encountering false positives. The reason for selecting FDR=0.1 as the threshold is to facilitate the detection of the genes with moderate changes during cell cycle. Often the case, we do believe that the multiple-testing correction is a very stringent approach, which is particularly suited when we want to identify the differentially expressed genes with high confidence in terms of per gene base. But for the generic pathway/functional related analysis, we could relax this stringency.

To unbiasedly present the result with different stringency in cutoff, we reanalyzed the DE genes with FDR < 0.05 and added the correspond data into the supplementary materials (Supplementary text 3).

Overall, 77.1% (1021 of 1325) of exon-based DE genes and 77.5% (746 of 962) of intron-based DE genes passed the criteria. We also examined the effects of applying a more stringent FDR cutoff on each type of CCGs. As a result, for Type I CCGs, 85.1% of them (332 of 390) passed exon FDR < 0.05 and 86.7% of them (338 of 390) pass intron FDR < 0.05; 73.7 % (689 of 935) of type II CCGs and 71.3% (408 of 572) of type III CCGs passed the more stringent FDR at exon or intron level respectively.

182-184 it is interesting to see such a large proportion of genes missed by existing analyses. I would like to see some more supporting information that these are not false positives.

To address this question, as suggested by Reviewer #4, we examined whether the novel CCGs identified in our data also exhibit cell cycle dependent changes in VASA-plate data. Despite the slight differences in culture conditions between VASA-seq study and ours (e.g., DMEM-F12 vs. DMEM, with antibiotics vs. without antibiotics), the novel CCGs identified by our methods also demonstrate cell-cycle dependent gene expression changes in the data generated by VASA-plate (Supp. Fig. 14). In addition, the expression patterns along cell cycle of these genes were well correlated between VASA-plate data and our data (Supp. Fig. 14). These observations indicate that the majority of the novel CCGs identified in our study are true.

Also, are there any canonical cell cycle genes, that are present in the data, but not detected by your analysis?

We thank the reviewer for this question, similar question has been raised by Reviewer #2. There are canonical cell cycle genes that were not captured by our analysis. Firstly, the genes regulated at the protein level along the cell cycle are not detected by our analysis. For example, cyclin D (CCND1, CCND2, CCND3) which drives G1/S transition, is tightly regulated by ubiquitin-proteasomal pathway along cell cycle^{2,3}, and no significant changes at the RNA level were detected in our data. Similarly, the activity of CDK4/6 is activated through the assembly with

cyclin D and is inhibited by the binding of inhibitory proteins from INK4 family or CIP/KIP family⁴⁻⁶. Consistently, no significant changes at the RNA level were detected in our data for these two genes.

Secondly, previous studies have shown that the regulation of cell cycle in different cell types requires different regulators^{5,7,8}. Therefore, there is a chance that some canonical cell cycle genes curated from previous studies may not participate in the cell cycle regulation in the cell line we studied here. Vice versa, some of the novel CCGs identified in our data could belong to the cell-type specific regulators of cell cycle.

201-203 as mentioned previously, include some cross comparison with other non parametric approaches.

As described above, we have employed GENIE3, a non-parametric approach, for co-variation analysis, and contrasted the results with those obtained from LASSO regression in the revised manuscript. The detailed results and the comparison are included in Supp. Fig. 11 & 17.

220- 222 I fully support the inclusion of other modalities.

However, it is well known that slight experimental changes can induce significant modifications in analyses.

You could use e.g. SCENIC on the RNA expression, infer some regulons, and then compare and contrast the results using the publicly available ChIP data.

We thank the reviewer for supporting the analysis with multiple modalities. In the manuscript, we have employed a similar strategy as suggested. We have also cited SCENIC as an alternative approach in the revised manuscript.

To explain our strategy in details, we firstly inferred the regulons based on RNA expression. The inferred regulons were then subjected to motif enrichment analysis similar to the scheme in SCENIC, and the enrichment analysis using the publicly available ChIP data. The results of motif enrichment analysis and the results using ChIP data were then compared and summarized in Supp. Table S4 and Supp. Table S9.

In most of cases, the regulons showing significant enrichment with ChIP data also exhibit significant enrichment of the corresponding motif. Exceptions were mainly observed in the cell cycle data, including the regulons driven by KLF11, MYBL2 and FOXM1 (Supp. Table S4). The underlying reason could be that these TFs do not have a consensus binding motif. For example, it has been reported that MYBL2 and FOXM1 form a complex to regulate the gene expression in G2 phase¹⁴, and in particular, FOXM1 binds directly to non-consensus sequences through the recruitment by cofactors¹⁵.

Interestingly, we also noticed that in senescence data, several regulons exhibit significant enrichment in motif analysis, but not in ChIP-based analysis. This could be caused by the different experimental conditions or simply the lack of ChIP-seq data for the corresponding TFs.

Therefore, we concluded that the motif-based analysis and the ChIP-based analysis could be complementary approaches to each other, although a more systematic evaluation is desired in future studies.

264-266 clarify what you mean by cross correlation analysis

Cross-correlation analysis is to measure the similarity of two time-series data sets (the expression of spliced RNA and unspliced RNA along cell cycle) as a function of the displacement of one relative to the other. If the gene expression is primarily driven by transcriptional regulation, we expect that the changes in spliced RNA would follow the changes in unspliced RNA. As a result, we would obtain a high coefficient between these two expression curve after moving one of them on the time series axis with the offset corresponding to the time delay between unspliced RNA and spliced RNA. Conversely, if the gene expression is not mainly driven by transcriptional regulation, we will not achieve any high coefficient between these two expression curves no matter how to shift the position of one curve.

324 - 336 it is unclear why you suddenly change dataset.

Even though the manuscript is presented as technical, some of the partial biological conclusions confuse the message. Moreover, it is unclear why some of analyses steps described for the first dataset are recapitulated on the second dataset (345-393).

If possible, I recommend focusing on one dataset, exemplifying the characteristics of the assay from all required angles, in a focused manner.

We thank the reviewer for this comment. We have tried to provide more information for the second part of analysis in the revised manuscript. We initially only want to use the regulation of cell cycle as a benchmarking model since it is believed to be well characterized. However, our result shows that cell cycle itself is a very complex dynamic process, with many gene expression changes along the cell cycle being unclear, which exemplifies the discovery mode of snapTotal method.

We agree with the reviewer that the paper is a bit overloaded. But considering the other two reviewers did not suggest removing the oncogene-induced senescence part of data, we decided to be conservative here and keep this part of study in the revised manuscript.

Minor comments

The main figures were not clearly labelled making it difficult to follow the information. I recommend presenting the code used to generate these results in a github repository. Moreover, if apps could be created to illustrate the dataset(s), it would be very useful for the

community.

We thank the reviewer for the suggestions here. We have improved the labeling of the main figures and deposited the related scripts on the Github.

Reviewer #4 (Remarks to the Author):

The manuscript entitled "Single-cell Total-RNA Profiling Unveils Regulatory Hubs of Transcription Factors" introduces snapTotal-seq, a promising addition to the growing pool of single-cell total RNA sequencing methods. The method was benchmarked against multiple single-cell RNA-seq assays for kinetic analysis of the cell cycle. The manuscript applies LASSO regression to discern transcription factor regulatory hubs that orchestrate cell cycle dynamics. Importantly, snapTotal-seq's ability to quantify total RNA at a single-cell level is a significant contribution to single-cell transcriptomic methodologies. I recommend publishing this manuscript with a few improvements to substantiate its findings and enhance reproducibility.

We thank the reviewer for his/her time and comments below.

1. The manuscript asserts that snapTotal-seq concurrently quantifies nascent and mature RNA. However, it is crucial to delineate that "nascent" RNA refers to transcripts undergoing synthesis. The RNA species captured by this method and analyzed in this manuscript are unspliced and spliced RNA. Given that intron retention and detention are known regulatory phenomena, it is essential to differentiate and accurately label unspliced or "intronic" RNA from "nascent" RNA. Throughout the manuscript, the term "nascent" should be replaced with either "unspliced" or "intronic".

We appreciate the reviewer's comment for improving the accuracy of our description. We have replaced the term 'nascent' with 'unspliced' or 'intronic' in the revised manuscript.

2. For a robust assessment of snapTotal-seq, I recommend a more quantitative approach in the benchmarking analyses. Clarity on the number of cells analyzed, selection criteria for cells, thresholds and parameters for cell inclusion, number of cells in Figure 1c, the proportion of intronic read and improvement in unspliced RNA capture efficiency over polyA-capture methods, and the distribution of read coverage across gene loci would provide a clear picture of the method's capabilities.

We thank the reviewer for the suggestions here. As suggested, for the analysis in Figure 1c, we have included the number of cells analyzed on the top of the figure in the revised manuscript. As all the cells we sequenced passed the initial quality control, we included all of them in this analysis. We added these statements to METHODS in the revised manuscript.

As suggested by the reviewer, we also examined the proportion of intronic reads and gene body coverage of our method. In summary, on average, $38.3 \pm 5.2\%$ (mean \pm sd) of reads were mapped to intron in our data, which is significantly higher than both polyA-capture methods.

Our data also demonstrated a more uniform read distribution along the gene body, in comparison to CEL-Seq2 and Smart-seq3. The mild 3' end bias observed in our data is likely due to the sequencing strategy we used here. Specifically, with short read sequencing (76 cycles in

total), we only used read 1 for mapping. As a result, for the fragments generated from the 5' end of transcript, only the inner part was sequenced.

In addition, we thoroughly investigated the capture efficiency of spliced and unspliced transcripts by different methods. The details are given in the response to Point #3 below. These new results are included in Sup. Fig. 5c-d in the revised manuscript.

3. A quantitative comparison of snapTotal-seq's performance against other single-cell RNA-seq methodologies in terms of dynamic range and bias towards RNA species would further improve clarity. Similarly, evaluation of the non-coding RNA, such as the proportion of reads from regulatory regions and whether snapTotal-seq has a higher capture efficacy for long RNA over short, would help in gauging whether the method truly and unbiasedly captures all RNA.

We would like to thank the reviewer for his or her thoughtful suggestions here. In the revised manuscript, we provided the detailed examination of the transcript capture efficiency across different RNA biotypes, dynamic ranges and RNA lengths.

Firstly, our method consistently demonstrated a higher gene detection rate across different RNA biotypes than both polyA-capture methods (Supp. Fig. 5e).

Secondly, in comparison to CEL-Seq2 and Smart-seq3, both our method and VASA-plate exhibit significant improvement in the capturing the transcripts with moderate or low expression levels at both exon and intron levels (Supp. Fig. 5f).

At last, both our method and VASA-plate exhibit a minor bias towards long RNA at both exon and intron levels, whereas polyA-capture based methods exhibit a clear bias towards long RNA at the intron level (Supp. Fig. 5g).

It is worth noting that although both total-RNA methods have a bias towards long RNA simply due the longer length, the overall capture efficiency for short RNA is still clearly higher than the capture efficiency of polyA-based methods for the same class of RNA. These results are included in Supp. Fig. 5 in the revised manuscript.

4. The choice of nucleotides for annealing and reverse transcription is surprising. Would a widely used random hexamer (NNNNNN), as opposed to T12, T3, and G3 sequences used in this method, mitigate potential read biases, particularly towards the 3'-end of genes?

We thank the reviewer for this comment. First, we include T12 in our primer mix to the purpose to capture the transcripts with poly A tails. In terms of potential biases related to T3 and G3 versus random hexamer, we would like to emphasize that with annealing at extremely low temperature, the unspecific binding of the common GAT handlers is the main determinant factor of which part of RNA will be primed from reverse transcription. The T3 and G3 sequence

were designed to increase initiation of reverse transcription. This beneficial effect of T3 and G3 versus NNNNNN has been shown previously in the original MALBAC paper for single-cell whole genome amplification.

5. The number of cells in other assays used in benchmarking was relatively low (100 – 200). Is this the total number of cells in these assays, or a small number of cells were randomly sampled to match the number of cells in snapTotal-seq? If it was sampled, what was the threshold/criteria?

The number of cells in other assays in the benchmark analysis are similar to our sampling size. In terms of criteria, we did not improvise a new one for other studies. The low-quality cells (e.g., with low UMI number) were filtered out by the same criteria as used in the original papers.

6. Concerning cell cycle analysis, the process for deriving five gene modules in Figures 2b, 4a, and 5f remains unclear. It would be beneficial to elucidate the clustering algorithm employed and provide module gene counts.

We apologize for our description in previous version. The gene modules were derived by using k-means clustering algorithm, and we include the detailed procedures in the caption in the revised manuscript. The gene counts in each module are also provided in the corresponding panels now in the revised manuscript.

7. The authors validated known CCGs and novel CCGs for Type I genes in Fig. 2f and 2g. Similar qRT-PCR-based validation for a handful of Type II and Type III genes would strengthen the robustness of CCGs classification. Such validation becomes further critical in the context of the claim that 80-90% of Type II and Type III genes do not have known cell cycle functions (Fig. 2h).

We thank the reviewer for this suggestion. As suggested by the reviewer here, we have randomly chosen 4 Type II genes and 4 Type III genes for qPCR-based validation. In brief, for the four type II genes, as shown in Supp. Fig. 13 in the revised manuscript, we observed significant changes at the spliced RNA level across different cell cycle stages, whereas only slight changes were observed at the unspliced RNA level. For the four type III genes, significant changes at the unspliced RNA levels were observed, but only slight changes were observed at the spliced RNA level. These new qRT-PCR has provided additional validation of our sequencing results.

8. It is essential to clarify whether the novel CCGs detected are unique to snapTotal-seq or simply overlooked or undetectable by other total-RNA-capturing single-cell methods. The authors could bolster the superiority of total-RNA-based methods by showing evidence of similar cell-cycle dependence of these novel CCGs in other single-cell total-RNA methods.

We would like to thank the reviewer for the suggestions here. As suggested, we have conducted additional analysis to show that the novel CCGs identified by our method also demonstrate cell-cycle dependent gene expression changes in the dataset generated by VASA-plate, and the expression patterns are well correlated between two datasets. As stated by the reviewer, these results indeed can bolster the superiority of total-RNA-based methods. The new analysis are included in Supp. Fig. 14 and additional information is provided in Supplementary text 1 in the revised manuscript.

9. The authors presented the post-transcriptional regulatory mechanisms governing Type II genes. However, I am not convinced that the evidence provided in Figure 4h substantiates the claim of a correlation between the expression patterns of four RNA-binding proteins and their respective target gene groups throughout the cell cycle. Should the RBP act to stabilize its bound RNA, one would anticipate a mirrored expression pattern between the target unspliced RNA and the RBP. Conversely, if the RBP serves to destabilize the RNA, their expression patterns should be inversely related. The Figure 4h does not seem to provide this evidence.

We thank the reviewer for the thoughtful comments here. In fact, as the reviewer stated, the effects does depend on whether the RBPs stabilize RNAs or destabilize the RNAs. Thus from the four RBPs, BCLAF1 and G3BP1 were known to enhance RNA stability^{16,17}. As a result, we successfully identified the target gene groups showing a positive correlation with the expression patterns of these two RBPs (Figure 4h). In contrast, PUM1 has been reported to promote RNA decay¹⁸⁻²⁰, as a result, we also identified its target gene groups whose gene expression patterns are negatively correlated with its own expression pattern (Figure 4h).

One explanation for the target gene groups lacking significant correlation with their corresponding RBPs is due to the false positives in the binding sites enrichment analysis, since the publicly available eCLIP datasets from ENCODE project were generated using different cell lines. Alternatively, besides these known RBPs, other regulatory mechanisms might also be involved in the regulation of these genes, potentially masking the correlation between their expression patterns and the ones of related RBPs. We have modified the corresponding figures and descriptions in the revised manuscript to clarify the limitation of this analysis.

10. Lastly, a thorough exposition of the LASSO regression model, including parameter choices and model validation, would enhance interpretability.

We thank the reviewer for the insightful suggestion. Indeed, the efficacy of using LASSO regression to capture the TF-gene links could depend on the selection of input TFs for analysis; for example, the inclusion of numerous unrelated TFs could introduce substantial noise, potentially leading to model failure.

Thus in the revised manuscript, we systematically evaluated the effects on the selection of input TFs on the results. Overall, we observed that the majority of TF-gene links can be robustly

captured with the parameters within the appropriate interval. The details are included in the Supplementary text 2 and Supp. Fig. 21.

On a minor note, the current layout of figures, with their numerous panels, tends to obscure the central findings. Transferring some of these panels to the supplementary material would help maintain focus on the most significant results.

We appreciated the suggestions here. We have gone through the organization of figures. To avoid major changes of main figures, we only reduced the panels in Figure 4 since this figure is particularly overcrowded. Specifically, we have removed panels k and l of Figure 4 into supplementary materials (Supp. Fig. 15c-d).

Reference

- 1 Niu, M. *et al.* Droplet-based transcriptome profiling of individual synapses. *Nat Biotechnol*, doi:10.1038/s41587-022-01635-1 (2023).
- 2 Diehl, J. A., Zindy, F. & Sherr, C. J. Inhibition of cyclin D1 phosphorylation on threonine-286 prevents its rapid degradation via the ubiquitin-proteasome pathway. *Genes Dev* **11**, 957-972, doi:10.1101/gad.11.8.957 (1997).
- 3 Diehl, J. A., Cheng, M., Roussel, M. F. & Sherr, C. J. Glycogen synthase kinase-3beta regulates cyclin D1 proteolysis and subcellular localization. *Genes Dev* **12**, 3499-3511, doi:10.1101/gad.12.22.3499 (1998).
- 4 Sherr, C. J. & Roberts, J. M. CDK inhibitors: positive and negative regulators of G1-phase progression. *Genes Dev* **13**, 1501-1512, doi:10.1101/gad.13.12.1501 (1999).
- 5 Malumbres, M. & Barbacid, M. Cell cycle, CDKs and cancer: a changing paradigm. *Nat Rev Cancer* **9**, 153-166, doi:10.1038/nrc2602 (2009).
- 6 Goel, S., DeCristo, M. J., McAllister, S. S. & Zhao, J. J. CDK4/6 Inhibition in Cancer: Beyond Cell Cycle Arrest. *Trends Cell Biol* **28**, 911-925, doi:10.1016/j.tcb.2018.07.002 (2018).
- 7 Malumbres, M. *et al.* Mammalian cells cycle without the D-type cyclin-dependent kinases Cdk4 and Cdk6. *Cell* **118**, 493-504, doi:10.1016/j.cell.2004.08.002 (2004).
- 8 Rane, S. G. *et al.* Loss of Cdk4 expression causes insulin-deficient diabetes and Cdk4 activation results in beta-islet cell hyperplasia. *Nat Genet* **22**, 44-52, doi:10.1038/8751 (1999).
- 9 Richart, L. *et al.* BPTF is required for c-MYC transcriptional activity and in vivo tumorigenesis. *Nat Commun* **7**, 10153, doi:10.1038/ncomms10153 (2016).
- 10 Gao, Q. *et al.* Knockdown of RREB1 inhibits cell proliferation via enhanced p16 expression in gastric cancer. *Cell Cycle* **20**, 2465-2475, doi:10.1080/15384101.2021.1987676 (2021).
- 11 Chen, D., Hinkley, C. S., Henry, R. W. & Huang, S. TBP dynamics in living human cells: constitutive association of TBP with mitotic chromosomes. *Mol Biol Cell* **13**, 276-284, doi:10.1091/mbc.01-10-0523 (2002).
- 12 Iwasaki, O. *et al.* Interaction between TBP and Condensin Drives the Organization and Faithful Segregation of Mitotic Chromosomes. *Mol Cell* **59**, 755-767, doi:10.1016/j.molcel.2015.07.007 (2015).
- 13 Teves, S. S. *et al.* A stable mode of bookmarking by TBP recruits RNA polymerase II to mitotic chromosomes. *Elife* **7**, doi:10.7554/eLife.35621 (2018).
- 14 Fischer, M., Schade, A. E., Branigan, T. B., Muller, G. A. & DeCaprio, J. A. Coordinating gene expression during the cell cycle. *Trends Biochem Sci* **47**, 1009-1022, doi:10.1016/j.tibs.2022.06.007 (2022).
- 15 Sanders, D. A. *et al.* FOXM1 binds directly to non-consensus sequences in the human genome. *Genome Biol* **16**, 130, doi:10.1186/s13059-015-0696-z (2015).
- 16 Savage, K. I. *et al.* Identification of a BRCA1-mRNA splicing complex required for efficient DNA repair and maintenance of genomic stability. *Mol Cell* **54**, 445-459, doi:10.1016/j.molcel.2014.03.021 (2014).

- 17 He, X., Yuan, J. & Wang, Y. G3BP1 binds to guanine quadruplexes in mRNAs to modulate their stabilities. *Nucleic Acids Res* **49**, 11323-11336, doi:10.1093/nar/gkab873 (2021).
- 18 Van Etten, J. *et al.* Human Pumilio proteins recruit multiple deadenylases to efficiently repress messenger RNAs. *J Biol Chem* **287**, 36370-36383, doi:10.1074/jbc.M112.373522 (2012).
- 19 Weidmann, C. A., Raynard, N. A., Blewett, N. H., Van Etten, J. & Goldstrohm, A. C. The RNA binding domain of Pumilio antagonizes poly-adenosine binding protein and accelerates deadenylation. *RNA* **20**, 1298-1319, doi:10.1261/rna.046029.114 (2014).
- 20 Goldstrohm, A. C., Hook, B. A., Seay, D. J. & Wickens, M. PUF proteins bind Pop2p to regulate messenger RNAs. *Nat Struct Mol Biol* **13**, 533-539, doi:10.1038/nsmb1100 (2006).

Reviewers' Comments:

Reviewer #2:

Remarks to the Author:

I appreciate the effort made by the authors in addressing all my points.

One last suggestion regards Supp. Fig 3, where the authors show the log₁₀ of number of genes detected per biotype. Could they also include a boxplot showing average number of reads (after normalization, I imagine)?

Reviewer #3:

None

Reviewer #4:

Remarks to the Author:

The authors have responded to the reviewers comment with appropriate revisions and comments. I congratulate them for this important paper.

Reviewer #5:

Remarks to the Author:

Blank

Reviewer #6:

Remarks to the Author:

I have thoroughly reviewed the authors' response to Reviewer 3's comments, and found that the authors have perfectly addressed all of Reviewer 3's suggestions. Therefore, this manuscript is suitable for publication.

In fact, the authors have addressed Reviewer 3's question about Smart-seq-total so well, that the rebuttal's figure (about Smart-seq-total heavily biasing toward RN7SK, RMRP, RPS29, and KIAA0907) might deserve a place in the Supplement. However, this is entirely up to the authors.

Beyond Reviewer 3's comments, my only minor suggestion is in the first paragraph, "It is worth noting that the majority of scRNA-seq methods, including the high-throughput platform of 10x chromium, are mature RNA only (oligo-dT) based methods," change "mature RNA only" to something like "primarily mature RNA". This will add rigor because later in Results, the authors acknowledged that these existing methods do accidentally capture a fraction of unspliced RNA ("the capture of unspliced RNA by dT-based methods relies on the binding of oligo-dT primer to the polyA sequence in introns").

Below please find our responses to the few suggestions from the reviewers. We highlight our replies in red fonts.

Reviewer #2 (Remarks to the Author):

I appreciate the effort made by the authors in addressing all my points.

One last suggestion regards Supp. Fig 3, where the authors show the log10 of number of genes detected per biotype. Could they also include a boxplot showing average number of reads (after normalization, I imagine)?

We thank the reviewer's comments here. To address the reviewer's question here, we have added the suggested boxplots showing the average UMI count detected by each method for each biotype in Supplementary Fig. 5f in the revised manuscript.

Reviewer #4 (Remarks to the Author):

The authors have responded to the reviewers comment with appropriate revisions and comments. I congratulate them for this important paper.

We thank this reviewer's comments and time for reviewing our paper.

Reviewer #6 (Remarks to the Author):

I have thoroughly reviewed the authors' response to Reviewer 3's comments, and found that the authors have perfectly addressed all of Reviewer 3's suggestions. Therefore, this manuscript is suitable for publication.

We thank the reviewer's comments here.

In fact, the authors have addressed Reviewer 3's question about Smart-seq-total so well, that the rebuttal's figure (about Smart-seq-total heavily biasing toward RN7SK, RMRP, RPS29, and KIAA0907) might deserve a place in the Supplement. However, this is entirely up to the authors.

We appreciate the reviewer's suggestion. We have added the related figure as Supplementary Fig 5e in the revised manuscript.

Beyond Reviewer 3's comments, my only minor suggestion is in the first paragraph, "It is worth noting that the majority of scRNA-seq methods, including the high-throughput platform of 10x chromium, are mature RNA only (oligo-dT) based methods," change "mature RNA only" to something like "primarily mature RNA". This will add rigor because later in Results, the authors acknowledged that these existing methods do accidentally capture a fraction of unspliced RNA ("the capture of unspliced RNA by dT-based methods relies on the binding of oligo-dT primer to the polyA sequence in introns").

We appreciate the reviewer's suggestion. We have rephrased the related description in the Introduction.